# Visualizing cellular and tissue ultrastructure using Ten-fold Robust Expansion Microscopy (TREx)

**Hugo GJ Damstra[1], Boaz Mohar[2], Mark Eddison[2], Anna Akhmanova[1], Lukas C Kapitein[1]\*, Paul W Tillberg[2]\***

[1]Cell Biology, Neurobiology and Biophysics, Department of Biology, Faculty of Science, Utrecht University, Utrecht, Netherlands; [2]Janelia Research Campus, HHMI, Ashburn, United States

**\*For correspondence:**
l.kapitein@uu.nl (LCK);
tillbergp@janelia.hhmi.org (PWT)

**Competing interest:** The authors declare that no competing interests exist.

**Abstract** Expansion microscopy (ExM) is a powerful technique to overcome the diffraction limit of light microscopy that can be applied in both tissues and cells. In ExM, samples are embedded in a swellable polymer gel to physically expand the sample and isotropically increase resolution in x, y, and z. The maximum resolution increase is limited by the expansion factor of the gel, which is four-fold for the original ExM protocol. Variations on the original ExM method have been reported that allow for greater expansion factors but at the cost of ease of adoption or versatility. Here, we systematically explore the ExM recipe space and present a novel method termed Ten-fold Robust Expansion Microscopy (TREx) that, like the original ExM method, requires no specialized equipment or procedures. We demonstrate that TREx gels expand 10-fold, can be handled easily, and can be applied to both thick mouse brain tissue sections and cultured human cells enabling high-resolution subcellular imaging with a single expansion step. Furthermore, we show that TREx can provide ultra-structural context to subcellular protein localization by combining antibody-stained samples with off-the-shelf small-molecule stains for both total protein and membranes.

## Editor's evaluation

The new robust Ten-fold Robust Expansion Microscopy method developed by the authors should be of wide interest to the cell biology community.

## Introduction

Expansion microscopy (ExM) circumvents the diffraction limit of light microscopy by physically expanding the specimen four-fold in each dimension (*Chen et al., 2015*; *Tillberg et al., 2016*). Expansion is achieved by chemically anchoring proteins and other biomolecules directly to a hyper-swelling gel, followed by aggressive proteolysis to enable uniform swelling of the gel material. While other super-resolution approaches are not readily compatible with thick tissue slices and require specialized optics (*Hell and Wichmann, 1994*), fluorophores (*Rust et al., 2006*), or software (*Gustafsson, 2000*), ExM is compatible with any microscope (*Gao et al., 2019*; *Zhang et al., 2016*), including other super-resolution modalities (*Gao et al., 2018*; *Halpern et al., 2017*; *Xu et al., 2019*), and performs well in both cultured cells and thick tissue slices (*Chen et al., 2015*; *Tillberg et al., 2016*). Assuming sufficiently high labeling density, the resolution increase of ExM depends on the expansion factor of the gel recipe used. Recently, ExM variants have been described that seek to improve resolution by increasing the expansion factor. For example, iterative ExM (iExM) uses sequential embedding in multiple expansion gels to achieve 15× and greater expansion but requires a complex sequence

of gel re-embedding, link cleaving, and fluorophore transfer (*Chang et al., 2017*), limiting its broad adoption.

The expansion factor of the gel itself can be improved by decreasing the concentration of crosslinker (*Okay, 2009*), usually bisacrylamide (bis), although this is generally at the expense of the mechanical integrity of the gel. For example, reducing the bis concentration in the original ExM recipe from 1.5 to 0.25 mg/mL produces an approximately nine-fold expanding gel (*Chen et al., 2015*, SF5), but these gels are too soft to hold their shape under the force of gravity. As a result, they are difficult to handle without breaking and display nonuniform expansion. This tradeoff of expansion versus gel mechanical integrity has not been explored in a quantitative or systematic way.

Another gel recipe variant, using a high concentration of the monomer dimethylacrylamide (DMAA), has enough crosslinking through side reactions and polymer chain entanglement that the crosslinker can be omitted entirely, producing ~10-fold expansion in one step (*Truckenbrodt et al., 2018*). This recipe has been used to expand cultured cells and thin cryosectioned tissue (*Truckenbrodt et al., 2019*), but reportedly requires rigorous degassing to remove oxygen prior to gelation, making it laborious to use. Moreover, expansion of thick tissue slices (>50 μm) has not been demonstrated using this method. Thus, a robustly validated and easily adoptable method that is compatible with multiple sample types and enables single-step expansion well over 4× without compromising gel integrity is lacking.

Here, we explored the expansion gel recipe space in a systematic manner, assessing the limits of single-round expansion using reagents and methods that would be familiar to labs already performing ExM. For any given choice of recipe parameters (monomer concentrations, gelation temperature, initiator concentration, etc.), varying the crosslinker alone yielded a family of recipes whose expansion factor and mechanical quality vary smoothly from high-expanding, mechanically unstable to low-expanding, tough gels. A range of crosslinker concentrations was tested for each family because the optimal crosslinker concentration may vary by family. From this exploration, we generated Tenfold Robust Expansion Microscopy (TREx), an optimized ExM method that allows for robust 10-fold expansion in a single step. We show that TREx can be used to expand both thick tissue slices and adherent cells. It is compatible with antibodies and off-the-shelf small-molecule stains for total protein and membranes. Together, we show that TREx enables 3D nanoscopic imaging of specific structures stained with antibodies in combination with cellular ultrastructure.

## Results

To systematically explore the expansion recipe space, we developed a streamlined approach for synthesizing dozens of gel recipes and characterizing their mechanical quality in parallel. For every set of gel recipe parameters (component concentrations and gelation temperature, listed in *Figure 1A*), we define a recipe family as the set of recipes generated by varying the crosslinker (bisacrylamide) concentration. For each family, we tested five recipes with crosslinker concentrations log-spaced from 1000 to 10 μg/mL, plus one with zero crosslinker. We also included the original ExM recipe, with 0.15% (1500 μg/mL) crosslinker. For each recipe, we cast three gel specimens, expanded them fully in water, and measured the expansion factor (*Figure 1B*). We found that resistance to deformation under the force of gravity was a good proxy for the more subjective judgment of ease of gel handling. We measured gel deformation by placing a semicircular punch from each gel upright on its curved edge and allowing the gel corners to deflect under the force of gravity. We defined the deformation index as the vertical displacement of each gel corner, divided by the gel radius (*Figure 1C*), which ranges from 0 (for gels that do not deform) to 1 (for gels that deform freely under their own weight). We manually calibrated this measurement, finding that deformation indices between 0 and 0.125 corresponded to gels with excellent ease of handling, 0.125–0.25 corresponded to acceptable ease of handling, and anything higher than 0.25 was unacceptable. While not as theoretically informative as elastic modulus and yield strength measurements, this measurement can be repeated and extended to new gel recipes by any lab developing expansion methods, without access to specialized equipment. We plotted the deformation index for each gel as a function of its expansion factor (*Figure 1D*) to directly assess the tradeoff between expansion and mechanical quality.

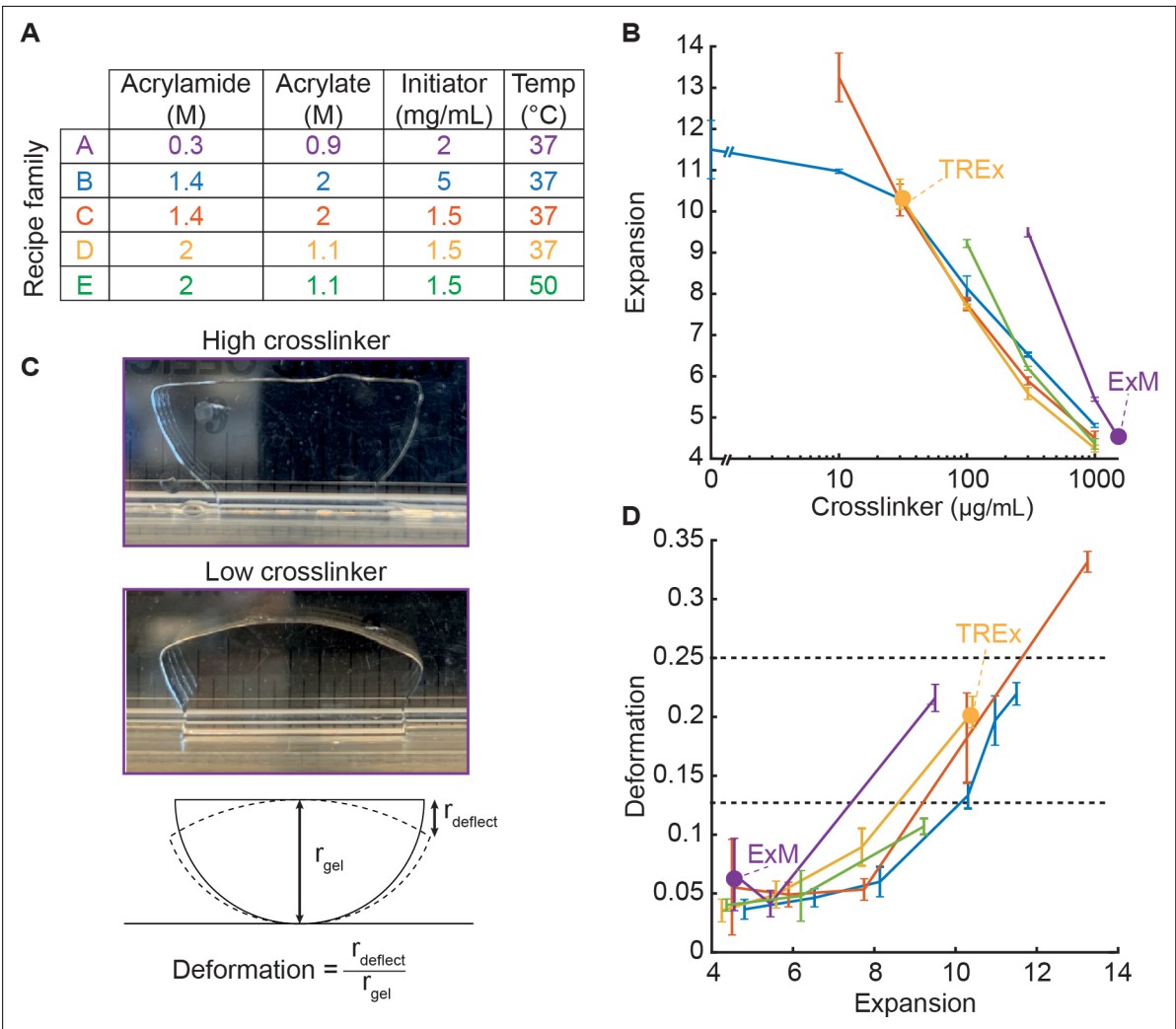

**Figure 1.** Development of Ten-fold Robust Expansion Microscopy (TREx) gel recipe. (**A**) Parameters of gel recipe families explored, including component concentrations and gelation temperature. Each family was characterized by keeping these conditions constant while systematically varying the crosslinker concentration. (**B**) Expansion factor (mean ± SD, n = 3) versus crosslinker concentration (log scale) for each gel recipe family without biological specimens. Line colors correspond to recipe families as in (**A**). Specific recipes are indicated with a filled purple dot (original expansion microscopy [ExM] recipe) and yellow dot (TREx). All recipe families were tested with crosslinker concentrations of 0, 10, 30, 100, 300, and 1000 µg/mL, plus an additional condition for family A with 1500 µg/mL, corresponding to the original ExM recipe. Only conditions in which gels formed are plotted. (**C**) Definition of gel deformation index. Example gels from recipe family A with high crosslinker and low deformation (top panel, 1.5 mg/mL), and low crosslinker and high deformation (middle panel, 300 µg/mL). Bottom panel: schematic illustrating deformation index measurement. (**D**) Deformation index (mean ± SD, n = 3) versus expansion factor for each gel recipe family without biological specimens, with line colors and dots corresponding to specific recipes as in (**A**) and (**B**). Horizontal gray lines indicate thresholds for gels with mechanical quality deemed perfect (deformation < 0.125) and acceptable (deformation < 0.25). Ideal recipes would occupy the lower-right quadrant, corresponding to high expansion and low deformability.

## Development of the TRex gel recipe

We began by characterizing a recipe family generated from the original ExM recipe (family A). Consistent with *Chen et al., 2015*, reducing the crosslinker to 300 µg/mL increased the expansion to ~9×, below which the gels fail to form consistently (*Figure 1B*, purple). We next characterized a high-monomer recipe family (family B) inspired by the 4×-expanding Ultra-ExM recipe from *Gambarotto et al., 2019*, which found that a higher monomer concentration relative to the original ExM recipe was necessary for high-fidelity preservation of the shape of centrioles. This was offset with a lower crosslinker (bisacrylamide) concentration of 0.1% (1000 µg/mL) to achieve 4× expansion. Indeed, for this high-monomer family of recipes, expansion as a function of crosslinker concentration was shifted leftward compared to standard ExM (*Figure 1B*, blue). As the crosslinker was decreased below 30 µg/

mL, the increase in expansion factor saturated around 11.5×. The deformation index versus expansion factor curve for the high-monomer family ran below that for standard ExM, indicating that for a given expansion factor the high-monomer gel holds its shape better than the corresponding standard ExM gel (*Figure 1D*, blue).

Compared with standard ExM, this high-monomer family uses a higher concentration of radical initiator and accelerator to trigger polymerization (5 mg/mL each of APS and TEMED, versus 2 mg/mL in standard ExM). We found that this high initiation rate causes gels to form within minutes at room temperature (RT). Because the rates of initiation and polymerization increase with temperature, it is likely that specimens are not fully equilibrating to the gelation temperature of 37°C before the onset of gelation, introducing a potential source of experimental variability. This rapid gelation makes the gelation chamber assembly step more time sensitive and presents challenges for adapting the technique to thick tissue slices as thick tissue slices require extra time for the gelation solution to diffuse throughout the sample prior to polymerization. Therefore, we also tested the same high-monomer recipe family but with initiator and accelerator reduced to 1.5 mg/mL (family C). The expansion versus crosslinker curve for this family was similar to family B for high crosslinker concentrations but displayed a slightly greater slope. Unlike family B, the expansion factor did not saturate upon decreasing the crosslinker concentration. Instead, the expansion factor continued to increase to 13× expansion at 10 μg/mL crosslinker (*Figure 1B*, red), with zero-crosslinker gels failing to form. Family C enables 10-fold expansion without sacrificing acceptable gel mechanical quality (30 μg/mL bis, *Figure 1D*, red), and without the faster, less controlled gelation kinetics of family B.

The recipe families explored above (B, C) feature a high fraction of sodium acrylate relative to acrylamide. Acrylate drives expansion of the gel but comes in widely varying purity levels and, in some cases, causes tissue to shrink. This macroscopic tissue shrinkage is modest compared to the gel expansion but may not be uniform at all scales. We therefore tested an alternative recipe family (D) with higher acrylamide to acrylate ratio (2.1:1). Increasing the acrylamide to acrylate ratio did not change the expansion factors appreciably at a given crosslinker concentration, suggesting that the swelling effect of acrylate saturates at high concentrations. At the maximum expansion factor of ~10, the deformation behavior was comparable to family C. We chose to proceed with family D due to its lower acrylate content.

We further tested an elevated gelation temperature of 50°C (family E) in an attempt to increase the initiation rate without introducing premature gelation as seen in recipe family B. Compared to family D, the expansion factors were around 15% higher at 100 μg/mL (6×) and 300 μg/mL crosslinker (9×), but gelation failed at lower concentrations, leaving family D as the family with a higher maximum expansion factor (i.e., 10× at 30 μg/mL bis). The deformation versus expansion curve for family E was similar to the other high-monomer recipe families (*Figure 1D*, green), but was found to be sensitive to processing details, such as the gelation chamber construction and placement within the incubator. This suggests that premature gelation prior to equilibrating at the higher temperature reduces the replicability of this recipe family.

Considering all five recipe families, family B (high acrylate and high APS/TEMED) displayed the lowest deformation index for a given expansion factor. Family D (high acrylamide and low APS/TEMED) displayed similar performance, with the deformation index remaining well within the acceptable range for expansion factors up to 10. In handling high-expanding (>8×) gels from all recipe families, we found that while those from the standard ExM family (A) were extremely prone to fragmentation, those from any of the high-monomer families could be handled more easily (and even dropped from a height of several feet) without breaking. Because the reduced initiator concentration of family D results in a slower and more controlled polymerization rate, and because we preferred a lower acrylate content, we chose this recipe family to proceed to biological specimen expansion. We found that the exact expansion factor varied for different specimens and gelation chamber geometries but could readily be adjusted by fine-tuning the crosslinker concentration. We thus recommend that users test gels with a range of crosslinker concentrations between 30 and 100 μg/mL to find a suitable recipe for their specimen preparation. We name the resulting method Ten-fold Robust Expansion (TREx) microscopy.

## Subcellular imaging of specific proteins and cellular ultrastructure in thick brain slices

In electron microscopy, nonspecific stains for proteins and membranes are commonly used to provide structural detail at high spatial resolution. Recently, the use of nonspecific NHS ester protein stains and other small-molecule probes has been combined with ExM (*M'Saad and Bewersdorf, 2020*; *Mao et al., 2020*; *Sim et al., 2021*; *Yu et al., 2020*). Expansion allows visualization of intracellular detail in such densely stained samples, which would otherwise be too crowded to lead to meaningful contrast. These applications have the promise to bring together the advantages of light microscopy (specific staining using antibodies and volumetric imaging) with the advantage of seeing cellular context typically provided by electron microscopy. Because TREx reaches single-step expansion factors at which small-molecule stains are expected to be useful, we set out to explore this idea further.

We applied the inexpensive, green-emitting dye BODIPY-FL NHS (total protein stain; see Materials and methods) after expansion with TREx to demonstrate total protein distribution in thick (100 μm) slices of mouse brain cortex (*Figure 2A*, *Figure 2—figure supplement 1*, *Figure 2—video 1*). The neuropil region outside the cell somas contained a rich profusion of fibers and structures visible in sharp relief. The nucleus of each cell was easily identified, with especially strong staining in nucleoli-like structures. Surrounding each nucleus, the nuclear envelope could be identified, with particularly dense total protein stain on the side facing the nucleus. The nuclear envelope was punctuated by heavily stained spots that span the envelope, consistent with nuclear pore complexes (NPCs; *Figure 2A*, inset). Within the cytosol, several organelles were marked by either heavy inner staining with a dim border or weak inner staining.

We attempted to optimize protein retention, according to the total protein stain intensity, by reducing both protein anchoring and proteolysis compared with the original ExM. We tested a range of anchoring strengths by varying the concentration of the acryloyl-X SE (AcX) anchoring molecule applied prior to gelation. This was done in combination with two reduced disruption methods: proteinase K applied at one-tenth that of the original ExM method (*Figure 2—figure supplement 1A*, top row) and a high-temperature, protease-free denaturation treatment (*Gambarotto et al., 2019*; *Ku et al., 2016*; *Tillberg et al., 2016*; *Zwettler et al., 2020*) similar to that employed in Western blotting (*Figure 2—figure supplement 1A*, bottom row). The protease-free treatment enabled greater protein retention but at the cost of incomplete expansion. This could be offset through reduced AcX concentration, though this was not clearly superior to high AcX followed by proteolysis, indicating a general tradeoff between protein retention and gel expansion (see Materials and methods). We chose a hybrid approach with moderate AcX anchoring and low-concentration proteinase K digestion followed by high-temperature denaturation to proceed.

We next tested whether antibodies, applied to the tissue using a standard immunofluorescence procedure before embedding, were also retained in the TREx gel. We stained mouse brain cortex tissue for Bassoon (a marker for both excitatory and inhibitory presynaptic active zones), Homer (a marker for the excitatory postsynaptic apparatus), and VGAT (a vesicular GABA transporter in the presynaptic compartments of inhibitory synapses). After staining and anchoring with AcX, tissue was expanded with TREx and imaged by light sheet microscopy. Numerous putative excitatory synapses were observed at high density, with clearly separated Bassoon and Homer pre- and postsynaptic staining (*Figure 2B and C*, *Figure 2—figure supplement 1B*, *Figure 2—video 2*). Because of the excellent axial resolution, TREx allowed us to quantify the separation of Bassoon and Homer in 3D, regardless of the angle of the synapse with respect to the imaging plane (*Figure 2D*). We found an average separation of 1.17 μm ± 0.52 μm (mean ± SD, 583 synapses), which, when corrected for expansion, is consistent with previous reports in cultured neurons that estimated the synapse separation between 90 and 130 nm (*Glebov et al., 2016*; *Wiesner et al., 2020*). Compared with Bassoon and Homer, VGAT had a more extended staining pattern, consistent with the known distribution of synaptic vesicles throughout presynaptic boutons. Elaborately shaped compartments with dense VGAT staining were seen with multiple synaptic release sites marked by Bassoon (*Figure 2C*). As expected, these release sites were not paired with the excitatory postsynaptic marker Homer. In addition, spatial correlation was observed in excitatory synapses between Homer and Bassoon, reflecting the transsynaptic alignment reported previously (*Hruska et al., 2018*; *Figure 2—figure supplement 1B*). These results demonstrate the ability of TREx to preserve correct synaptic staining while enabling sub-diffraction limited imaging of large tissue sections.

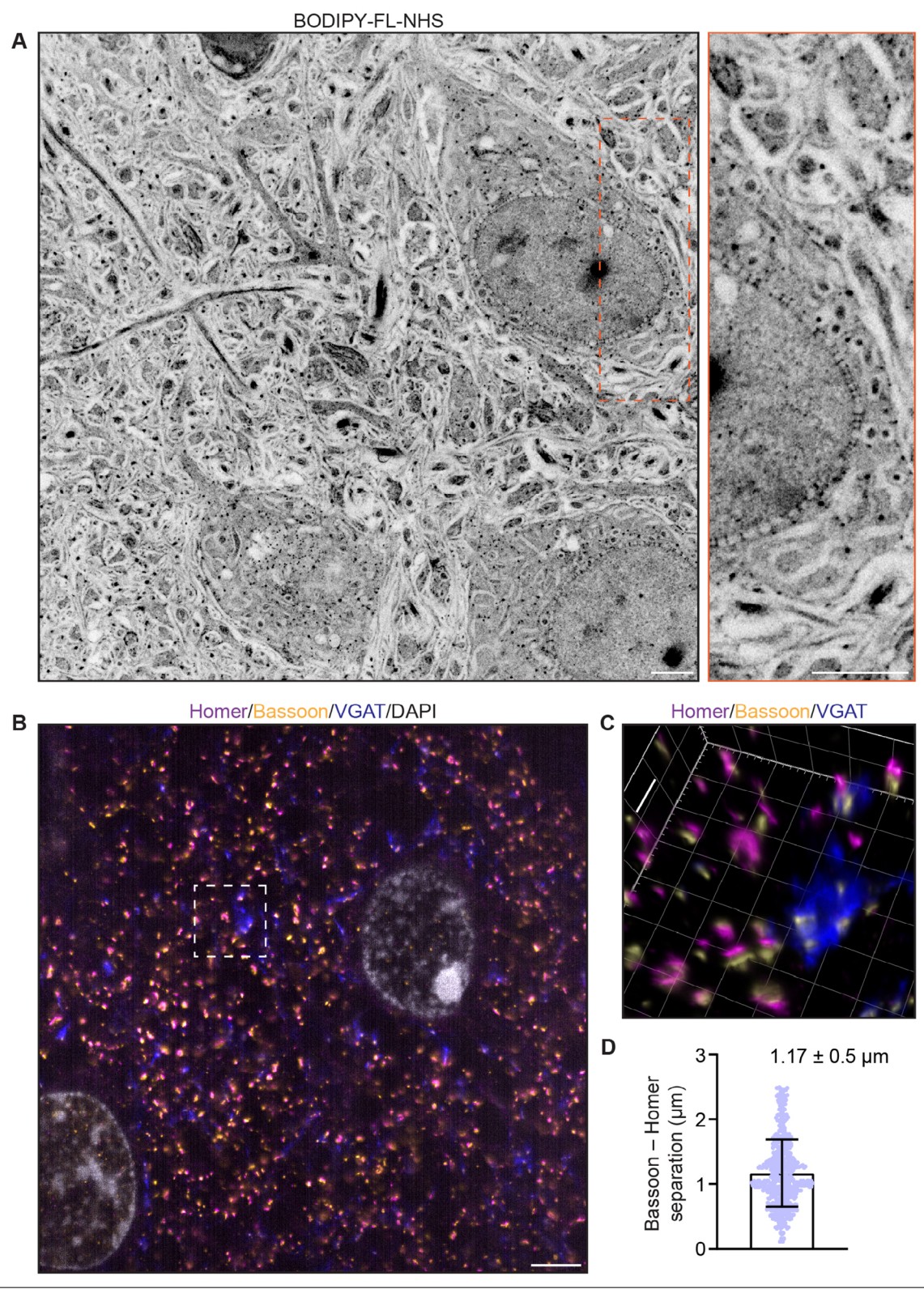

**Figure 2.** Ten-fold Robust Expansion Microscopy (TREx) in mouse brain tissue slices. (**A**) Mouse brain tissue (cortex) expanded using TREx, stained for total protein content with BODPIY-FL NHS, and imaged by confocal microscopy. Displayed contrast is inverted to show dense stained regions as dark. Inset: zoom-in showing nuclear envelope with densely stained structures spanning the nuclear envelope, consistent with nuclear pore complexes. (**B**) Mouse brain tissue (cortex) stained with antibodies against Homer (magenta), Bassoon (yellow), VGAT (blue), and DAPI (gray), and expanded using

*Figure 2 continued on next page*

*Figure 2 continued*

TREx. (**C**) Volumetrically rendered zoom-in of white box in (**A**) showing paired Bassoon- and Homer-rich structures, consistent with excitatory synapses. Depending on the orientation, clear separation of Bassoon and Homer can be observed, as well as a complex, structured presynaptic vesicle pool marked by VGAT bearing several release sites marked by Bassoon. (**D**) Quantification of Bassoon and Homer separation (mean ± SD plotted, n = 538 synapses, one replicate). Scale bars (corrected to indicate pre-expansion dimensions): main ~2 μm, zooms ~400 nm.

The online version of this article includes the following video and figure supplement(s) for figure 2:

**Figure supplement 1.** Comparison of anchoring and disruption conditions.

**Figure 2—video 1.** Z-stack of Figure 2A.

https://elifesciences.org/articles/73775/figures#fig2video1

**Figure 2—video 2.** 3D render of Figure 2B.

https://elifesciences.org/articles/73775/figures#fig2video2

## Validation of expansion factor and deformation

Increasing the expansion factor from 4 to 10× could result in greater sensitivity of the expansion factor to local variation, for example, in protein dense complexes, resulting in less uniform expansion. To examine this, we explored the nanoscale isotropy of TREx by imaging NPCs, which have a highly stereotyped and well-characterized structure. The NPC constituent protein nucleoporin 96 (NUP96) appears in a ring structure with a 107 nm diameter (*Thevathasan et al., 2019*). NPCs have recently been explored as a reference structure for super-resolution microscopy methods, including ExM in combination with other super-resolution methods (*Pesce et al., 2019*; *Thevathasan et al., 2019*). For the conventional 4–5× expansion approach, this revealed that the diameter of the NPC was 14–29% smaller than expected from the macroscopic expansion of the gel. We used the NUP96-GFP homozygous knock-in cell line from *Thevathasan et al., 2019* to study the quality of nuclear pore expansion using TREx with well-validated anti-GFP antibodies (*Figure 3A*). After expansion with TREx, individual NPCs were uniformly retained and clearly visible using diffraction-limited confocal microscopy (*Figure 3B*). An antibody against NUP153 similarly demonstrated individual NPCs but with less complete NPC coverage compared with the antibody stain against the NUP96-GFP tag (*Figure 3C*). The macroscopic gel expansion factor was 9.5×, suggesting an expected NPC size after expansion of 107 nm × 9.5 = 1.02 μm. We used a semi-automated approach to determining the diameter of 60 NPCs randomly chosen from three nonadjacent cells and found the size after expansion to be 939 nm ± 90 nm (mean ± SD) (*Figure 3D*). This is about 8% smaller than expected based on the macroscopic expansion of the gel and implies a local expansion factor of 8.8×, or 92% of the expected 9.5×. These data indicate that TREx offers more uniform local expansion compared to conventional ExM.

Other reference structures for super-resolution microscopy include microtubules and clathrin-coated pits (CCPs), which have been extensively studied with various modalities including ExM (*Bates et al., 2007*; *Chen et al., 2015*). We fixed cells and stained for both tubulin and clathrin heavy chain and quantified the diameter of CCPs (*Figure 3E and F*). We found an average diameter of 1.16 ± 0.2 μm (mean ± SD), which, corrected for 10-fold expansion, is consistent with previously reported values (*Chen et al., 2015*; *Jones et al., 2011*). In samples with sufficiently high labeling densities (e.g., the periphery of COS7 cells), we could furthermore resolve the stereotyped M-profile across peripheral microtubules in COS7 cells, indicative of the hollow structure of microtubules. As expected for microtubules stained with primary and secondary antibodies (*Mikhaylova et al., 2015*), we observed a 450-nm peak-to-peak distance after expansion (*Figure 3G*).

Next, we quantified the measurement error introduced by nonuniform expansion by comparing antibody-stained microtubules imaged before expansion with 3D gSTED versus after expansion with confocal microscopy (*Figure 3H–J*), as described previously (*Chen et al., 2015*). Measurement lengths between pairs of points after expansion were compared to the distance expected given uniform expansion, and the average fractional deviation plotted as a function of measurement length (*Figure 3I*). For a large tiled acquisition of 42 fields of view from one gel (~650 × 750 μm after expansion), the measurement error was found to be a constant fraction (3.2% ± 1.7%) of the measurement length (*Figure 3I*). We used the similarity transform to calculate the overall expansion factor of the entire imaged area and found it to be 9.4×, consistent with the expansion expected for the whole gel. Together, these data show that TREx enables uniform single-step, 10-fold expansion that retains

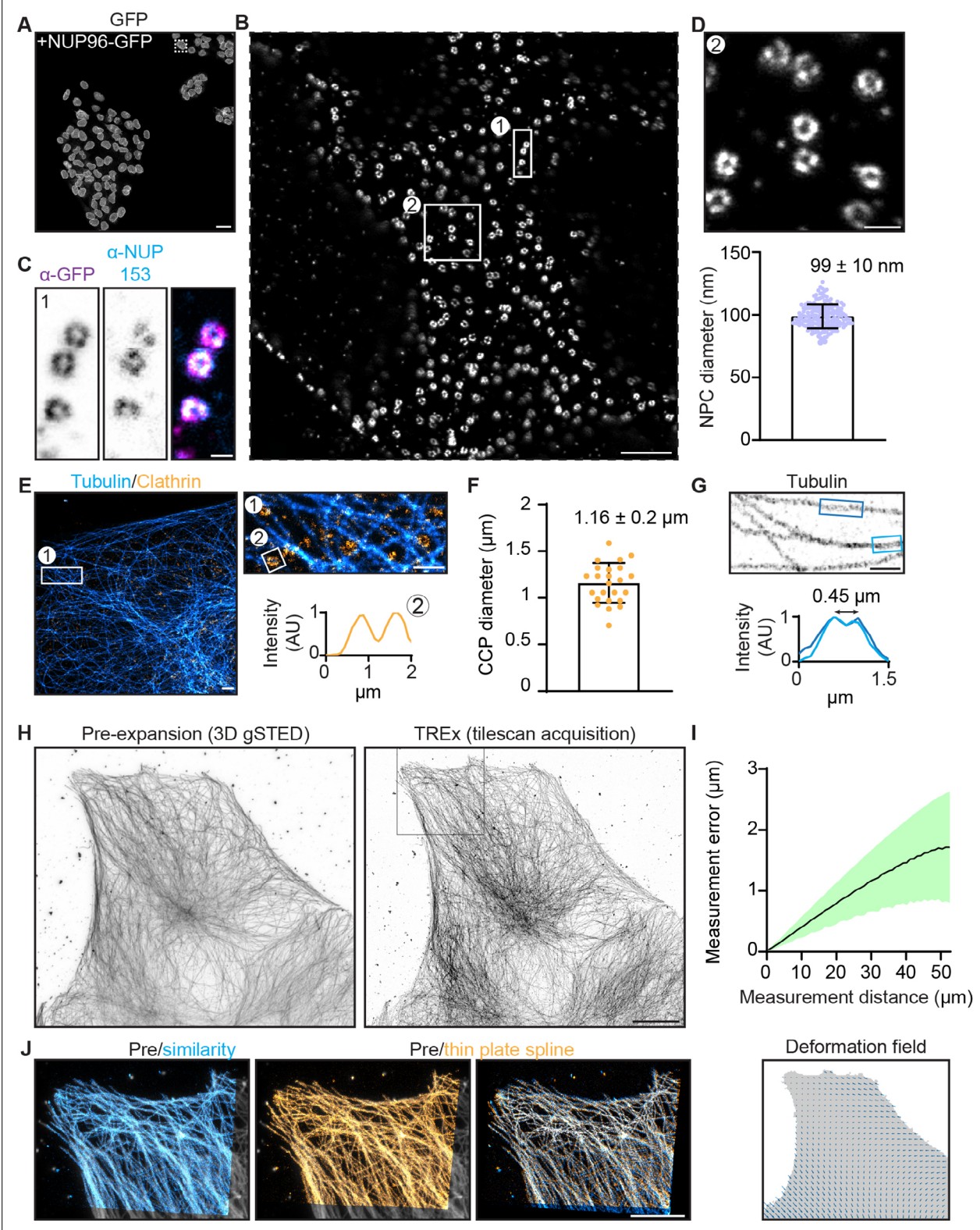

**Figure 3.** Characterization of expansion isotropy using Ten-fold Robust Expansion Microscopy (TREx). (**A**) U2OS knock-in cells with homozygous NUP96-GFP, amplified with anti-GFP antibodies. (**B**) One nucleus from boxed region of (**A**), imaged by confocal microscopy after TREx. (**C**) High-resolution view of several nuclear pores from boxed region (1) of panel (**B**), showing both anti-GFP (magenta) and anti-NUP153 (endogenous nuclear pore protein, cyan) staining. (**D**) High-resolution view of several nuclear pores from boxed region (2) of panel (**B**) (top). Distribution of diameters of individual nuclear pores (bottom), corrected for the macroscopic expansion factor of 9.5×. N = 60 nuclear pore complexes (NPCs) from three spatially separated cells. (**E**) U2OS

*Figure 3 continued on next page*

*Figure 3 continued*

cells stained for clathrin heavy chain and tubulin, representative line scan over clathrin-coated pit (CCP) showing central null. (**F**) Quantification of CCP diameter. Plotted mean ± SD (1.16 ± 0.2 µm) of 25 CCPs from five cells (two independent experiments). (**G**) High-resolution view of microtubules in extracted COS7 cell and corresponding line scans with mean peak-to-peak distance indicated. (**H**) Maximum projection of pre-expansion 3D gSTED acquisition (left) and maximum projection of tilescan acquisition (42 tiles, post-expansion size ~750 × 650 µm) of the same cell post-expansion (right). (**I**) Post-expansion single field of view, as indicated with magenta box in (**D**), aligned with the pre-expansion image (gray) by similarity transformation (cyan) or thin plate spline elastic transformation (orange). Right shows overlay of similarity and elastic transformation to illustrate local deformations. (**J**) Quantification of measurement errors of the stitched dataset due to nonuniform expansion. Mean error for a given measurement length (black line) ± SD (shaded region). The residual elastic deformation field is shown below. Scale bars (corrected to indicate pre-expansion dimensions): (**A**) 50 µm, (**B**) ~1 µm, (**C**) ~100 nm, (**D**) ~200 nm, (**E**) (overview) ~1 µm, zooms (**E**) and (**G**) ~500 nm, (**H**) ~10 µm, (**J**) ~5 µm.

structural detail over large distances, in both cultured cells and thick tissue slices, with similar performance compared with the original 4× ExM.

## TREx enables subcellular localization of proteins and cellular ultrastructure in cultured cells

We next explored the use of TREx for high-resolution imaging of specific proteins, NHS ester stains, and lipid membranes in cultured cells. For membranes, a custom-synthesized membrane probe compatible with the ExM process has been shown to visualize membranes in fixed brain tissue (*Karagiannis et al., 2019*). This probe relies on a peptide-modified lipid tail that intercalates in target membranes and provides opportunities for anchoring to the gel through D-lysines in its peptide sequence. We asked whether the commercially available membrane-binding probe mCLING could also be used for membrane staining and gel anchoring. mCLING has been developed as a fixable endocytosis marker consisting of a fluorophore and a short polypeptide group with one cysteine and seven lysines coupled to a palmitoyl membrane anchor (*Revelo and Rizzoli, 2016*). Due to the presence of multiple lysines, we hypothesized that mCLING would be compatible with standard ExM anchoring through AcX. While the standard protocol for mCLING delivery relies on active endocytosis in living cells, we tested whether mCLING would stain intracellular membranes more uniformly when added to fixed cells, which would have the added benefit of not perturbing intracellular membrane trafficking by long incubation in live cells. To test this, we fixed activated Jurkat T cells, incubated the fixed cells with mCLING overnight, and proceeded with the TREx protocol. We found that mCLING efficiently intercalates in both the plasma membrane and internal organelles and is retained following our standard anchoring procedure (*Figure 4A*, *Figure 4—video 1*).

By carefully rendering the imaged volumes, we could, with one probe, both appreciate the ruffled morphology of the plasma membrane on top of the flattened part of the cell and visualize the organelle clustering typical of activated T cells (*Figure 4A and B*). As in electron microscopy, where distinct morphologies are used to identify organelles, we could clearly identify different organelles based on mCLING, suggesting that it could be used for automated segmentation of organelles. Indeed, we found that mitochondria could be readily segmented using a trainable Weka segmentation algorithm (*Figure 4B*; *Arganda-Carreras et al., 2017*). While the resolution of subcellular structures is limited by the density of mCLING moieties in the membrane, the efficiency of crosslinking to the gel, and the maximum expansion factor, we found that TREx allows sufficient single-step expansion to resolve individual mitochondrial cristae (*Figure 4C*), which are known to be as closely spaced as 70 nm (*Stephan et al., 2019*). Although mCLING is membrane impermeable in live cells (due to multiple positively charged amino groups), it readily stained fixed and unpermeabilized cells following extended incubation. Because this approach does not require labeling live cells and is expected to reduce differences in uptake efficiency between intracellular compartments, we used this approach in all subsequent experiments.

We next tested if mCLING could also be used to visualize membranes in more complex cell types. To test this, we used differentiated Caco-2 cells grown to form an epithelial monolayer. Using TREx, we could expand the entire monolayer and visualize membranes using mCLING (*Figure 4D–H*, *Figure 4—video 2*). The advantage of optical, volumetric imaging is underscored by the fact that we can easily render one dataset in several ways, either resembling scanning electron microscopy to highlight volumetric surface morphology (*Figure 4D*) or transmission electron microscopy to explore single planes in more detail (*Figure 4E and F*). For example, we were able to resolve the elaborate interdigitated

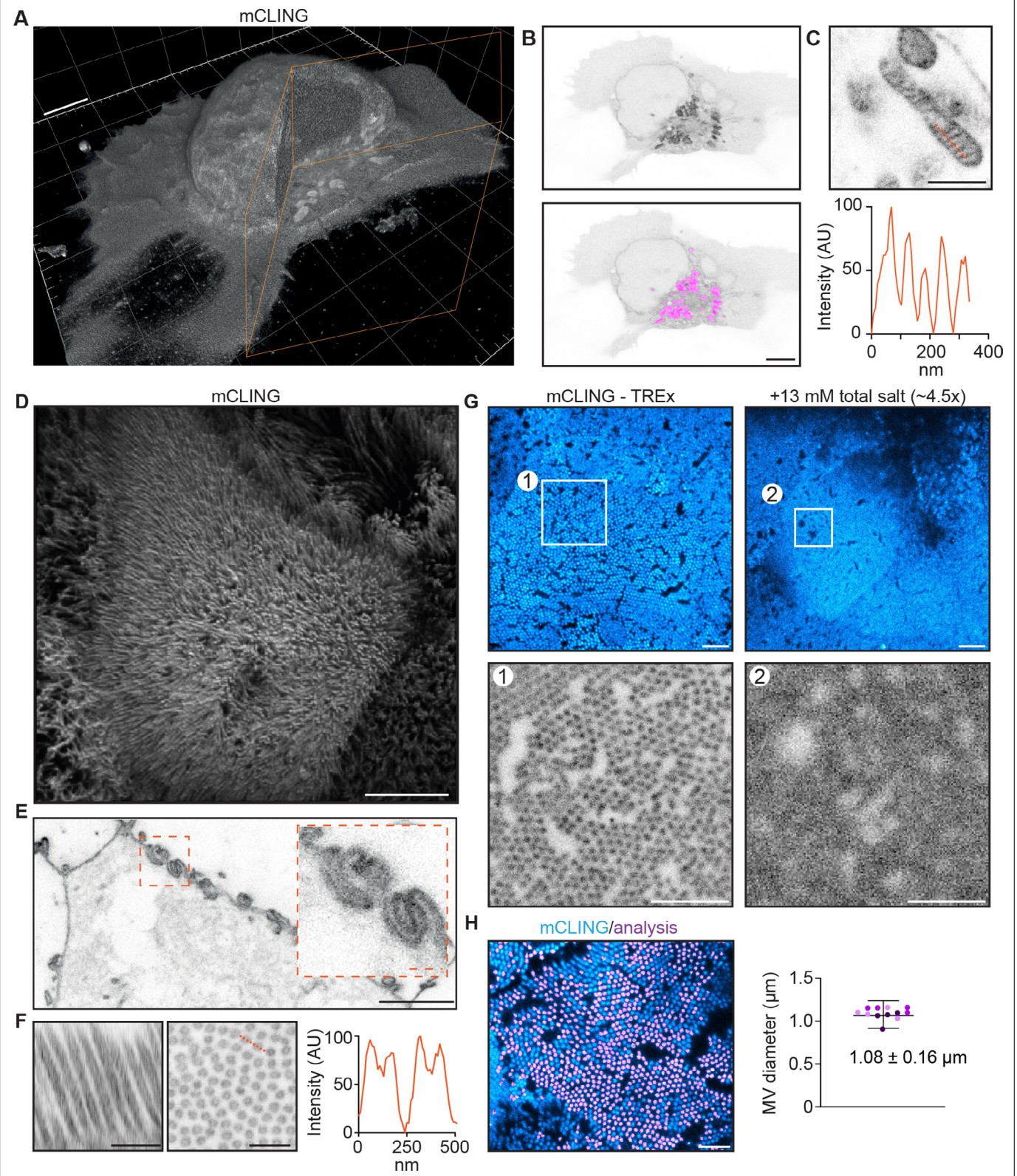

**Figure 4.** Ten-fold Robust Expansion Microscopy (TREx) can be used to visualize the ultrastructure of cellular membranes. (**A**) Volumetric render of Jurkat T cell activated on anti-CD3-coated coverslip fixed and stained using mCLING. Colored clipping planes indicate portion clipped out to reveal intracellular detail. (**B**) Immunological synapse of activated T cell in (**A**) revealing organelle clustering at the immunological synapse. Below: mitochondria segmented using the trainable Weka segmentation algorithm indicated in magenta. (**C**) Representative example of mitochondrion in

*Figure 4 continued on next page*

*Figure 4 continued*

T cells visualized with mCLING. Line profile along the orange dashed line indicates mitochondrial cristae. (**D**) Depth-coded volumetric projection of Caco2 monolayer apical brush border as seen from above looking down on the cells. (**E**) Representative plane below the apical surface revealing highly interdigitated cell-cell contacts. (**F**) Resliced (left) representative zoom (right) of brush border showing microvilli as hollow protrusions. Line scan indicated in orange. (**G**) Comparison of dense brush borders after 10-fold expansion in water (left) and ~4.5× expansion in 13 mM salt (right, see *Figure 4—figure supplement 1*). Single plane of brush border and plane of same cell below the apical surface shown in cyan. Zooms 1 and 2 correspond to areas of the same size corrected for the expansion factor to illustrate the increase in resolution of tenfold expansion. (**H**) Quantification of microvilli diameter by determining the area of cross-sectioned (left). Plotted mean ± SD (107.7 ± 16.1 nm) of 12,339 microvilli with means of individual cells color coded per replicate overlayed (four cells per replicate, N = 3). Scale bars (corrected to indicate pre-expansion dimensions): (**A, B, D, E**) (main) ~2 µm, (**C, E**) (zoom), (**F**) ~ 500 nm, (**G, H**) ~ 1 µm.

The online version of this article includes the following video and figure supplement(s) for figure 4:

**Figure supplement 1.** Expansion factor versus ionic strength.

**Figure 4—video 1.** 3D render of Figure 4A.

https://elifesciences.org/articles/73775/figures#fig4video1

**Figure 4—video 2.** 3D render of Figure 4D.

https://elifesciences.org/articles/73775/figures#fig4video2

cell-cell junctions that could previously only be clearly appreciated using electron microscopy (*Drenckhahn and Franz, 1986*), as well as resolve individual microvilli as hollow membrane protrusions within the dense brush border. To underscore the significant resolution increase of TREx compared to standard ExM, we incubated expanded TREx gels with solutions of increasing ionic strength to shrink the gel back to ~4.5× the size of the pre-expanded gel (*Figure 4G*, and *Figure 4—figure supplement 1*). When the 10× and 4.5× expanded gels were imaged, dense brush borders of differentiated cells could only be resolved in the 10× gel (*Figure 4G*). To validate the expansion factor, we quantified the diameter of individual microvilli as these have been thoroughly characterized with EM with a diameter of ~100 nm (*Crawley et al., 2014*). Indeed, we found an average diameter of 1.08 ± 0.16 µm (n = 12,339 from 12 cells, N = 3), which, corrected for an expansion factor of 10, is within 8% of the value from electron microscopy. Together, these data illustrate the robustness of TREx in expanding multiple cell types and show how the increased expansion factor combined with a commercially available membrane stain provides rapid volumetric insights into the elaborate membranous architecture of cells.

Previously, we used ExM to study the three-dimensional organization of microtubules in neurons and T cells (*Hooikaas et al., 2020*; *Jurriens et al., 2021*; *Katrukha et al., 2021*). In these experiments, cells were typically pre-extracted with detergent and glutaraldehyde to reduce background, followed by paraformaldehyde fixation (*Tas et al., 2017*). We reasoned that the increased expansion of TREx would dilute the soluble tubulin background, resulting in a relative boost in signal over background and eliminating the need for pre-extraction to enable high-resolution imaging of microtubules in combination with membranes. To test this, we fixed cells without pre-extraction, treated them with mCLING, stained for tubulin, and imaged the stained cells both before and after expansion with TREx (*Figure 5A*, second panel). Expanded cells retained high-quality anti-tubulin antibody signal exhibiting high contrast relative to the cytosolic background. We also observed increased detail in both mCLING and tubulin stains after expansion compared to before expansion, which was particularly apparent with side views of the same cell (*Figure 5A*, far right).

We next fixed cells expressing GFP-Sec61β without pre-extraction, treated them with mCLING, stained for GFP and tubulin, and then proceeded with TREx (*Figure 5B–D*, *Figure 5—video 1*). Because small-molecule stains like mCLING do not rely on subsequent antibody amplification we reasoned that this decrease in linkage error should be reflected in the apparent size of subcellular structures. To test this, we quantified the diameter of endoplasmic reticulum (ER) tubules that were stained by both mCLING and GFP-Sec61β (*Figure 5C*). Indeed, we observed an average tubule diameter of 0.79 ± 0.1 µm for mCLING, which, corrected for 10-fold expansion, is consistent with the previously published range of 60–100 nm (*Shibata et al., 2009*; *Shibata et al., 2006*), while the anti-GFP signal from the same tubules is broadened to an average diameter of 0.93 ± 0.1 µm. Within the same dataset, the interplay between microtubules and ER (*Figure 5D*) in three dimensions could be resolved, which revealed how other membranous organelles were connected to both structures

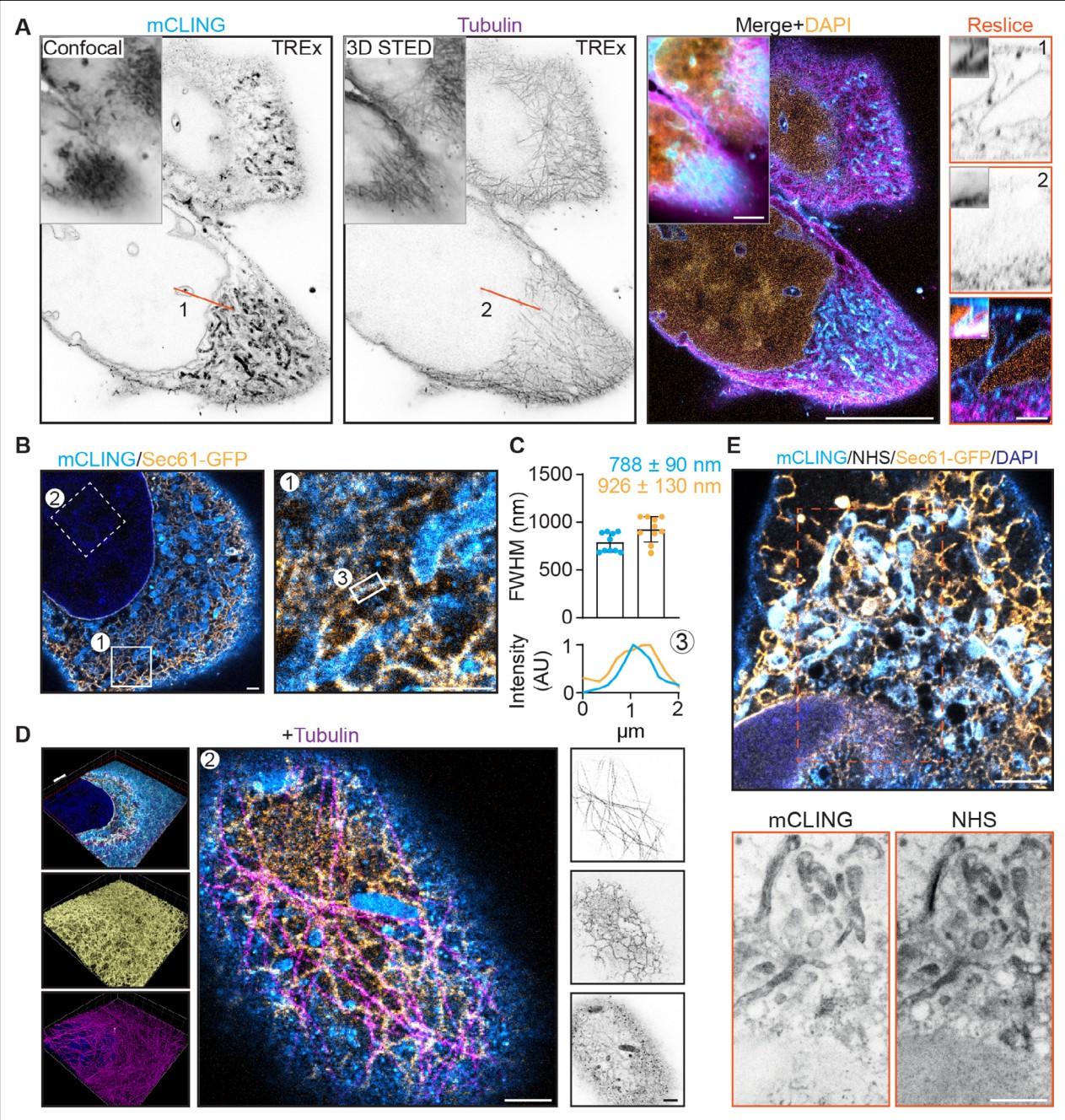

**Figure 5.** Ten-fold Robust Expansion Microscopy (TREx) microscopy can combine antibody-based staining with NHS ester total protein stain to provide subcellular context. (**A**) Single and merged planes of expanded U2OS cell stained for mCLING, tubulin, and DAPI; gray outlined insets show similar confocal and 3D STED acquisitions pre-expansion, for mCLING and tubulin, respectively. Single planes of mCLING and tubulin are displayed in inverted contrast. Orange line (1, 2) corresponds to reslices (left) with insets showing similar resliced planes pre-expansion. (**B**) U2OS cell expressing GFP-Sec61β stained for mCLING, GFP, and tubulin (tubulin channel not shown). (**C**) Quantification of endoplasmic reticulum (ER) tubule diameter of both mCLING and GFP-Sec61β channels. Plotted mean ± SD (787.7 ± 0.09 nm for mCLING and 925.5 ± 0.13 nm for GFP-Sec61β) full width at half maximum (FWHM) of 10 line scans over ER tubules positive for both mCLING and GFP-Sec61β from four cells (two independent experiments). Below: representative line scan of region indicated by (3) shown. (**D**) Left: volumetric render of cell in (**B**). Top portion of cell is clipped with clipping plane indicated in red. Volumetric render of entire volume for GFP and tubulin in inset (**A**) and (**B**), respectively. Middle: zoomed region of top of cell (indicated by box 2 in **B**). Right: single channels from middle panel displayed in inverted contrast revealing the tight spatial organization. (**E**) Merged plane of expanded U2OS cell expressing GFP-Sec61β stained for mCLING, GFP, NHS ester, and DAPI. Single planes of mCLING and NHS ester are displayed in inverted contrast. Scale bars (corrected to indicate pre-expansion dimensions): (**A**) (main) ~5 µm, (**D**) (renders) ~2 µm, (**A**) (reslices), (**B**) (single planes), (**D**) (single planes), (**E**) ~ 1 µm.

The online version of this article includes the following video for figure 5:

*Figure 5 continued on next page*

*Figure 5 continued*

**Figure 5—video 1.** 3D render of Figure 5B.

https://elifesciences.org/articles/73775/figures#fig5video1

(*Figure 5D*). Thus, TREx facilitates high-resolution three-dimensional mapping of specific cytoskeletal and membranous structures in combination with markers that provide ultrastructural context.

Finally, we tested whether TREx using general membrane stains could be combined with general protein stains and/or antibody stains. U2OS cells transfected with GFP-Sec61β were fixed, treated with mCLING, stained for GFP, and expanded with TREx followed by the NHS stain (*Figure 5E*). Because we performed the NHS stain after disruption, we used high-temperature denaturing disruption rather than proteolytic digestion. We found that secondary antibodies that had been used to stain GFP before gelation withstood this disruption step. We observed a clear degree of overlap between mCLING and NHS, especially in the dense perinuclear region, but we could also identify distinct features of each stain (*Figure 5E*, bottom). These results demonstrate that general stains for membranes and proteins can be combined with antibody-based labeling to reveal specific proteins in their ultrastructural context.

## Discussion

We developed TREx in order to expand biological specimens 10-fold in a single round of expansion, without specialized equipment or procedures. In developing this method, we established a framework for assessing gel recipes operating near this apparent limit of single-round expansion. We found that the mechanical performance of gel recipes, that is, resistance to deformation versus gel expansion factor, varies smoothly with changes in crosslinker. For all high-monomer (~3 molar total acrylamide and acrylate) gel recipe families, the relation between expansion factor and crosslinker concentration fell close to a common curve. The high radical initiation rate of family B enabled gelation without the inclusion of a crosslinker, suggesting that side reactions and polymer entanglement in these conditions create sufficient network crosslinks to form a gel. Gel deformation measurements plotted versus expansion factor, though less precise, also show high-monomer recipe families falling close to a common curve. Compared to the high-monomer families, the original ExM recipe family is less resistant to deformation for a given expansion factor and expands more for a given crosslinker concentration. The factor determining gel properties is not crosslinker concentration in the gel recipe per se, but rather the density of effective crosslinks formed between neighboring polymer chains in the gel. This suggests that the original, low-monomer recipe less efficiently incorporates crosslinker molecules as network crosslinks. This may be because the resulting lower rate of chain extension allows incorporated crosslinker molecules to be re-reacted by the same polymer chain before they can react with neighboring polymers.

While the expansion factor of the original ExM recipe can be tuned by varying the crosslinker concentration, it has been shown that increasing the monomer content is required to maintain nanoscale isotropy using centrioles as a convenient standard reference structure (*Gambarotto et al., 2019*). Considering gel quality versus expansion factor alone, the high-monomer recipe family B derived from the U-ExM recipe allows for a 10-fold expanding gel (at crosslinker concentration of 30 μg/mL) with a low deformation index of 0.13. However, the high radical initiation rate used in this family (5 mg/mL APS and TEMED) results in fast gelation. This increases the time sensitivity of mounting the specimen in the gelation chamber and adds an additional challenge for adapting the method to thick tissues, which require extended incubation in the gelation solution. Recipe families C and D solve this problem by reducing initiation rates at a slight expense of mechanical performance compared with family B. Like family B, family C has a high acrylate content, which might contribute to imaging artifacts due to shrinkage of the sample prior to gelation and inconsistent acrylate purity. Family D reduces the acrylate content by half while retaining similar mechanical performance to family C, especially in the 10-fold expansion regime. Finally, with family E we explored whether increasing the gelation temperature to 50°C would produce the improved mechanical performance of family B (through increased temperature-dependent radical initiation) without premature gelation at RT. However, we found that this reduced the expansion factor and increased susceptibility to experimental variation. Therefore, we based our TREx recipe on recipe

family D. The exact crosslinker concentration that produces 10-fold expanding gels can vary due to differences in gelation chamber design, so we recommend that each lab test a range of cross-linker concentrations between 30-60 ppm using their choice of specimen, gelation chamber, and incubator.

We applied TREx to mouse brain tissue slices stained either for specific targets with antibodies, or for total protein distribution with NHS ester dyes. Single-round 10-fold expansion with TREx followed by total protein staining was sufficient to reveal densely packed axons and dendrites running through the neuropil, while individual organelles could be resolved within the neuronal soma. The nuclear envelope, along with presumptive NPCs, was also clearly resolved. The correct relative localizations of pre- and postsynaptic markers and presynaptic neurotransmitter vesicles stained with standard immunofluorescence were also retained in TREx-expanded tissue slices. For this characterization, we used a hybrid disruption approach incorporating reduced proteinase K digestion followed by high-temperature denaturation. We adopted this hybrid approach because we had noticed that sometimes the reduced protease treatment alone produces underexpanded nuclei while the protease-free treatment alone produced underexpanded synapses. In general, a higher degree of anchoring requires higher disruption strength to achieve full expansion on the macroscopic level. Further application-specific optimization may be beneficial, given the heterogeneity of biological tissue. For applications where maximizing total protein retention is not a priority, we recommend simply using a high concentration of proteinase K (e.g., 1:100 dilution, overnight).

We validated the nanoscale isotropy of TREx by comparing the size of well-characterized subcellular structures to values from literature. These include NPCs, ER tubules, CCPs, and microvilli. In the case of NPCs, the conventional 4× expansion approach yielded an NPC diameter 14–29% smaller than expected (*Pesce et al., 2019*; *Thevathasan et al., 2019*), suggesting that protein-dense complexes may resist full expansion. Using TREx, the average NPC diameter was 8% smaller than the expected value. Further optimization of anchoring and disruption conditions may improve expansion uniformity for protein-dense structures such as NPCs. The widths of CCPs, ER tubules, and microvilli measured with TREx are reasonable, given the variation in previously measured sizes of these structures or different staining and imaging modalities. Notably, we observed the broadening of apparent ER tubule width due to antibody linkage error compared to direct membrane labeling with mCLING.

In addition to nanoscale isotropy, we characterized the overall expansion isotropy by comparing microtubules before and after expansion, finding expansion-induced measurement errors on average 3.2% of a given measurement length. Although these errors do not saturate at sub-100 µm distances as they do for lower-expanding gels, the nanoscale fidelity over short length scales is in line with previous expansion methods (*Chen et al., 2015*; *Tillberg et al., 2016*) and accumulation of nanoscale deformation over longer distances is not a limiting factor for most biological applications. For experiments that would require lower long-distance deformations, we expect these deformations could be reduced through optimized gel handling and mounting.

We further demonstrated the utility of TREx for the study of cell biology through combinations with several staining modalities in cultured cells, prepared in several culture formats. After a single round of expansion with TREx, the commercially available membrane stain mCLING was able to clearly resolve the internal structure of mitochondria and the detailed pattern of plasma membrane ruffling in activated Jurkat T cells. While these structures would be readily resolved with electron microscopy of mechanically sectioned cells, we were able to do so in the context of complete cells, enabling detection and automated segmentation of mitochondria clustered followed T cell activation. Caco-2 cells grown on permeable filters were also successfully stained with mCLING and expanded with TREx to reveal the detailed structure of epithelial microvilli and membrane interdigitations at the contacts of neighboring cells. These structures had previously been known from electron microscopy but were now imaged with ease in the context of entire cell monolayers. Successful application of TREx to filter-cultured Caco-2 cells further demonstrates the robustness of TREx because we had repeatedly failed to cleanly recover epithelial cultures using standard ExM. We speculate that this may be due to the increased robustness of TREx to interactions between the silicone filter material and the gelation process. This robustness is further demonstrated by its adoption in other biological systems (*Gros et al., 2021*) including in cultured neurons (*Ozkan et al., 2021*) and primary cultured human cells (*Nijenhuis et al., 2021*), and by its superior mechanical properties as measured by traditional materials characterization methods (*Chen et al., 2021*).

In U2OS cells, TREx retained anti-tubulin antibody stain with high efficiency, maintaining continuous microtubules with high signal-to-noise ratio after expansion. Moreover, we demonstrate that this specific labeling can be combined with total membrane labeling (mCLING) to reveal close appositions between microtubules and various organelles including ER and presumptive mitochondria. We also combined mCLING with total protein labeling (NHS ester dye), finding that the staining patterns were similar in their overall contours, albeit with some clear differences, such as the presence of presumptive NPCs in the NHS ester channel. It has been shown (*Sim et al., 2021*) that highly hydrophobic NHS ester dyes exhibit strong contrast for cytosolic organelles while highly hydrophilic NHS ester dyes strongly stain the nucleus. The moderate hydrophobicity dyes that we used (BODIPY-FL [*Zanetti-Domingues et al., 2013*] and Alexa Fluor 594 [*Hughes et al., 2014*]) exhibit both nuclear staining and contrast for cytosolic organelles. The strong overlap between NHS ester and mCLING stains is likely due to a combination of the reactivity of NHS esters towards unreacted lysines in the mCLING molecule and hydrophobic interactions that boost the reaction rate between the dye and proteins in membrane-rich organelles.

In summary, by systematically exploring the ExM recipe space, we established a novel recipe using standard ExM reagents that has been rapidly adopted by other labs. TREx allows for 10-fold expansion of both thick tissue slices and cells in a single expansion step and has applications in tissue and high-resolution subcellular imaging. Importantly, TREx of antibody-stained samples can be combined with off-the-shelf small-molecule stains for both membranes and total protein to localize specific proteins in their ultrastructural context.

# Materials and methods

**Key resources table**

| Reagent type (species) or resource | Designation | Source or reference | Identifiers | Additional information |
|---|---|---|---|---|
| Antibody | Goat polyclonal anti-rabbit IgG (H+L) Alexa Fluor 488 | Abcam | Cat# 150077; RRID:AB_2630356 | IF (1:500) |
| Antibody | Goat polyclonal anti-mouse IgG3 Alexa Fluor 594 | Invitrogen | Cat# A-21155; RRID:AB_2535785 | IF (1:500) |
| Antibody | Goat polyclonal anti-chicken CF633 | Biotium | Cat# 20126; RRID:AB_10852831 | IF (1:500) |
| Antibody | Goat polyclonal anti-rabbit IgG (H+L) Alexa Fluor 594 | Molecular Probes | Cat# A11037; RRID:AB_2534095 | IF (1:400–1:200) |
| Antibody | Goat polyclonal anti-mouse IgG (H+L) Alexa Fluor 488 | Molecular Probes | Cat# A11029; RRID:AB_2534088 | IF (1:400–1:200) |
| Antibody | Goat polyclonal anti-chicken IgY (H+L) Alexa Fluor 488 | Molecular Probes | Cat# A11039; RRID:AB_2534096 | IF (1:400–1:200) |
| Antibody | Mouse monoclonal anti-CD3 | STEMCELL Technologies | Cat# 60011 | IF (1:250) |
| Antibody | Rabbit monoclonal anti-α-tubulin | Abcam | Cat# ab52866; RRID:AB_869989 | IF (1:250) |
| Antibody | Mouse monoclonal anti-clathrin heavy chain | Thermo Fisher Scientific | Cat# MA1-065; RRID:AB_2083179 | IF (1:250) |
| Antibody | Chicken polyclonal anti-GFP | Aves Labs | Cat# GFP-1010; RRID:AB_2307313 | IF (1:400–1:200) |
| Antibody | Rabbit polyclonal anti-NUP153 | Abcam | Cat# ab84872; RRID:AB_1859766 | IF (1:200) |
| Antibody | Chicken polyclonal anti-Bassoon | Synaptic Systems | Cat# 141016; RRID:AB_2661779 | IF (1:300) |
| Antibody | Rabbit polyclonal anti-Homer | Abcam | Cat# 97593; RRID:AB_10681160 | IF (1:300) |
| Antibody | Mouse monoclonal IgG3 anti-VGAT | Synaptic Systems | Cat# 131011; RRID:AB_887872 | IF (1:300) |
| Cell line (*Homo sapiens*) | U-2 OS-CRISPR-NUP96-mEGFP | Cell Lines Service | Cat# 195 | |
| Transfected construct (*H. sapiens*) | pAc-GFPC1-Sec61beta | Addgene | Cat# 15108; RRID:Addgene_15108 | |

## Recipe space exploration

### Gelation chambers

A glass slide served as the bottom piece of each gelation chamber. Four strips of 250-µm-thick adhesive silicone material (DigiKey Cat# L37-3F-320-320-0.25-1A), ~ 3 mm wide and running the width of the slide, were adhered to the slide to partition it into three separate chambers, each ~12 mm wide. A plus-charged glass slide was placed over the silicone strips to form the top of the gelation chamber and held in place with tape. Two sides of each chamber were open to air, providing a convenient fill port for adding ~100 µL of monomer solution after chamber construction.

### Gel synthesis and characterization

Sodium acrylate was made by neutralizing acrylic acid (Sigma, 147230) with NaOH until the pH reached the range of 7.5–8. Initial neutralization (until pH ~7) was done with 10 N NaOH on ice and using a fume hood. Neutralization was done in a volume of water calculated to yield a final concentration of 4 M sodium acrylate. The gel recipes for each family contained 1× phosphate buffered saline (PBS) and the amounts of acrylamide (Sigma, A4058), sodium acrylate, and initiator (APS, Sigma, A3678) indicated in *Figure 1A*. Each gel recipe contained the same amount of TEMED (Sigma, T7024) as APS. For each recipe family, gelation solution with crosslinker withheld (but including APS and TEMED) was premixed on ice, in one tube for each recipe family. This solution was then split into six tubes and mixed with serial dilutions of crosslinker (bisacrylamide, Sigma, M1533) to yield complete gelation solution with final crosslinker concentrations (in µg/mL) of 1000, 300, 100, 30, 10, and 0. Complete gelation solution was pipetted into gelation chambers and incubated at 50°C for 1 hr (family E) or 37°C for 2 hr (families A–D). Gels were then cooled for 15 min at RT and chamber tops carefully removed. Gels typically remained stuck exclusively to the top (plus charged) slide. Samples of each gel were taken with a 6 mm biopsy punch, taking care to avoid material within ~2 mm of the chamber edges (to avoid oxygen exposure from air or silicone material during gelation). Excess gel was scraped away with a razor blade. A few drops of distilled water were pipetted onto each gel to help release them from the glass slide. Each 6 mm gel specimen was gently released from the slide with a razor blade, placed in a 9 cm Petri dish and expanded by washing with excess water 2 × 15 min followed by 2 × 1 hr. Diameters of expanded gels were measured and divided by 6 mm to obtain the expansion factor. A semi-circle 25 mm in diameter was punched from each gel using a cookie cutter. Semi-circular gel punches were placed in a plastic tray, which was stood up on end so that the gel stood upright on its curved side, allowing the flat edge to deform under the force of gravity. Each gel was photographed, with a ruler positioned for scale. Using ImageJ, each top edge was described by seven manually chosen points, which were then fit to a circle. This best-fit circle was used to calculate the vertical deviation of the gel corners, which was divided by the gel radius to obtain the deformation index.

### TREx gelation solution

Sodium acrylate was either purchased (Sigma, 408220) or made by neutralizing acrylic acid as described above. The TREx gelation solution contains 1.1 M sodium acrylate, 2.0 M acrylamide (AA), 50 ppm N,N'-methylenebisacrylamide (bis), PBS (1x), 1.5 ppt APS, 1.5 ppt TEMED, and (optionally, for thick tissue slices) 15 ppm 4-hydroxy TEMPO (4HT, Sigma, 176141).Monomer solution was made by combining all components of gelation solution except APS, TEMED, and 4HT. Monomer solution may be aliquoted and stored at –20°C, but must be thawed at RT and vortexed before use to redissolve any acrylamide crystals that may have precipitated at low temperature before freezing. Fully dissolved monomer solution may be kept on ice for up to several hours before crystallization occurs. 4HT, TEMED, and APS were added to monomer solution to produce gelation solution directly before use.

## Tissue experiments

### Fixation and antibody staining

Mice were transcardially perfused with ice-cold 4% formaldehyde in 100 mM sodium phosphate buffer, pH 7.4. Brains were dissected out and post-fixed in 4% formaldehyde at 4°C overnight (*Figure 2A*) or for 2 hr (*Figure 2B*), followed by washing with PBS (1×) and slicing by vibratome at 100 µm. For *Figure 2B*, slices were stained with standard IHC procedures. Primary antibodies were used at 1:300

dilution in PBS with 0.1% Triton and 2% bovine serum albumin (BSA) (PBT) overnight at 4°C (chicken anti-Bassoon, Synaptic Systems, Cat# 141016, RRID:AB_2661779; rabbit anti-Homer, Abcam, Cat# 97593, RRID:AB_10681160; mouse IgG3 anti-VGAT, Synaptic Systems, Cat# 131011, RRID:AB_887872). Sections were washed 3 × 30 min in PBT and stained for at least 6 hr in secondary antibodies 1:500 in PBT at RT (goat anti-rabbit Alexa Fluor 488, Abcam, Cat# 150077, RRID:AB_2630356; goat anti-mouse IgG3 Alexa Fluor 594, Invitrogen, Cat# A-21155, RRID:AB_2535785; goat anti-chicken CF633, Biotium, Cat# 20126, RRID:AB_10852831). Stained sections were washed 3 × 30 min in PBS.

### TREx

Brain slices were treated with 100 μg/mL (*Figure 2A*) or 10 μg/mL (*Figure 2B*) acryloyl-X SE (Thermo Fisher, A20770) in PBS (diluted from a 10 mg/mL anhydrous DMSO stock solution) for 1 hr at RT, followed by rinsing with PBS. Slices were incubated with TREx gelation solution (using 50 μg/mL bis and with 4HT added up to 15 μg/mL) for 20 min on ice with shaking. Following incubation on ice, each tissue specimen was placed on a glass slide at RT. Four dabs of vacuum grease were applied to the slide, with each dab at least several millimeters from the tissue specimen. A coverslip was placed over the tissue and vacuum grease dabs, and pressed down until contacting the tissue, taking care not to let the tissue slide around on the slide. The vacuum grease served to hold the assembly in place, thus forming the gelation chamber. Gelation solution was pipetted into the chamber from the side to fully surround the tissue. The chamber was incubated at 37°C for 1 hr to complete gelation. Following embedding, excess gel was removed with a razor blade, and gelled slices were recovered into PBS. The gel for *Figure 2B* was digested in proteinase K (NEB, P8107S) diluted 1:1000 in PBS for 3 hr at RT and washed in PBS 4 × 30 min. Gels for both *Figure 2A and B* were then placed into disruption buffer (5% SDS, 200 mM NaCl, 50 mM Tris pH 7.5) in a 2 mL Eppendorf tube and incubated at 80°C for 3 hr followed by rinsing in 0.4 M NaCl and washing 2 × 30 min in PBS. Gels were stained with BODIPY-FL NHS (total protein stain) at 10–20 μM (*Figure 2A*) or DAPI at 200 μg/L (*Figure 2B*) in PBS for 1 hr at RT. Gels were placed in glass-bottom six-well plates and washed in Milli-Q water 3 × 15 min followed by 2 × 1 hr to fully expand. Gels were imaged using a Zeiss LSM 800 confocal microscope with ×40/1.1 NA, water immersion objective (*Figure 2A*), or Zeiss Z1 lightsheet microscope with ×10/0.3 NA illumination objectives and ×20/1.0 NA water immersion detection objective (*Figure 2B*).

### Image processing

For *Figure 2A*, raw data was drift corrected using Huygens Professional (SVI) and imported into ImageJ, where a sum projection of two planes (z-spacing: 0.8 μm) was made. *Figure 2B* is a maximum projection of two planes (z-spacing: 0.38 μm) and indicated zoom is a volumetric render of the raw data in Arivis.

### Synaptic distance

Raw data was segmented using ilastik Pixel and Object segmentation workflows (*Berg et al., 2019*). For each Homer-positive segmented object (postsynaptic compartment), the closest Bassoon-positive segmented object (presynaptic compartment) was selected. Synaptic distance was defined as the distance between the local peaks in intensity that were closest to the mask center of mass in 3D.

### Synaptic marker intensity line plots

For *Figure 2—figure supplement 1B*, max projections of 31 planes in three fields of view were used to get the intensities of Bassoon and Homer from synapses that were perpendicular to the imaging plane. The values were then peak normalized.

## NPC experiment

### Cell culture, fixation, and antibody staining

U2OS cells with homozygous GFP-NUP96 knock-in (Cell Lines Service, no. 195) were maintained in DMEM (Corning) supplemented with 10% fetal bovine serum (FBS) (Gibco), 1% L-glutamine (Gibco), and 1% penicillin-streptomycin (Gibco). Cells were tested for mycoplasma contamination prior to use for this work. Exponentially growing cells were harvested and seeded onto 12 mm, No. 1 coverslips (Carolina Biological Supply) for use in ExM. Cells were grown at 37°C and 5% $CO_2$. Cells were fixed

with 4% formaldehyde (EMS, RT 15714) in 1× PBS for 10 min at RT, then rinsed with 1× PBS. Cells were stained with standard immunocytochemistry (ICC) procedures. Primary antibodies were used at 1:200 dilution in PBS with 0.1% Triton and 2% BSA (PBT) for 2 hr at RT (chicken anti-GFP, Aves Labs, Cat# GFP-1020, RRID:AB_10000240; rabbit anti-NUP153, Abcam, Cat# ab84872, RRID:AB_1859766), followed by washing 3 × 5 min in 1× PBS. Secondary antibodies were used at 1:200 dilution in PBT for 2 hr at RT or at 4°C overnight (goat anti-chicken Alexa Fluor 488, Thermo Fisher Scientific, Cat# A11039, RRID:AB_2534096; goat anti-rabbit Alexa Fluor 594, Thermo Fisher Scientific, Cat# A11037, RRID:AB_2534095), followed by washing 3 × 5 min in 1× PBS. Stained cells were imaged before expansion on an epifluorescence microscope, Nikon Ti-E with ×60/1.2 NA water immersion objective. The imaged region was indicated by marking the back of the coverslip with a marker.

### TREx

Fixed cells were anchored with 100 µg/mL AcX in 1× PBS for 1 hr at RT and embedded using the TREx gelation solution. The gelation chamber was constructed from a 20-mm-diameter, adhesive-backed silicone gasket (Sigma, GBL665504) affixed to a glass slide. The 12 mm coverslip with cultured cells was affixed to the center of the gelation chamber with a dab of vacuum grease and covered with PBS. TEMED and APS were then added to the TREx monomer solution on ice and mixed well to produce gelation solution. The PBS was tipped off from the cells, which were rinsed with ~100 µL of gelation solution. Approximately 200 µL of gelation solution was placed into the gelation chamber, which was sealed with a 22-mm-square #2 coverslip. The completed gelation chamber was placed at 37°C for 1 hr to complete gelation. The chamber was disassembled and the gel carefully trimmed with a curved scalpel into a right trapezoid shape centered around the pre-gelation imaged area. The trimmed trapezoid was photographed with a ruler for scale, quickly to avoid shrinking due to evaporation, and recovered into PBS. The gel was then digested with proteinase K (NEB, P8107S) diluted 1:1000 in PBS for 3 hr at RT and washed in PBS 4 × 30 min. Digested gels were placed into disruption buffer (5% SDS, 200 mM NaCl, 50 mM Tris pH 7.5) in a 2 mL Eppendorf tube and incubated at 80°C for 3 hr followed by rinsing in 0.4 M NaCl and washing 2 × 30 min in PBS. Disrupted gels were expanded fully with several washes in deionized water, photographed again with a ruler for scale, and imaged with a Zeiss LSM 800 confocal microscope with ×40/1.1 NA water immersion objective.

### Data analysis

The gel size before and after expansion was measured from the gel photographs. The centers of 60 randomly chosen NPCs in three nonadjacent cells were identified manually and saved as an ROI list in ImageJ. The radial intensity distribution of each NPC was computed using the 'Radial Profile Plot' plugin (https://imagej.nih.gov/ij/plugins/radial-profile.html) and saved as a .csv. Radial intensity distributions were loaded into MATLAB for further processing. A Gaussian distribution was fit to a window in the middle of each profile, and the center of the Gaussian was taken as the radius of the corresponding NPC.

## Wild-type, transfected, and T cell experiments

### Cell culture

Jurkat T cells (clone E6.1) were grown in RPMI 1640 medium w/ L-glutamine (Lonza) supplemented with 9% FBS and 1% penicillin/streptomycin. For T cell activation, 18 mm #1.5 coverslips (Marienfeld, 107032) were coated with poly-D-lysine (Thermo Fisher Scientific, A3890401), washed with PBS and incubated overnight at 4°C with a mouse monoclonal anti-CD3 (clone UCHT1, STEMCELL Technologies, #60011) 10 µg/mL in PBS. Cells were spun down for 4 min at 1000 rpm and resuspended in fresh, prewarmed RPMI 1640 medium, after which cells were incubated on the coated coverslips for 3 min prior to fixation. U2OS and COS7 cells were cultured in DMEM medium supplemented with 9% FBS and 1% penicillin/streptomycin. U2OS cells were transfected with GFP-Sec61β (Addgene, 15108) using FuGENE6 (Promega). Caco2-BBE cells (a gift from S.C.D. van IJzendoorn, University Medical Center Groningen, the Netherlands) were maintained in DMEM supplemented with 9% FBS, 50 µg/µL penicillin/streptomycin and 2 mM L-glutamine. For imaging, cells were seeded on 6.5 mm Transwell filters (3470; Corning) at a density of $1 \times 10^5$ /cm$^2$ and cultured for 10–12 days to allow for spontaneous polarization and brush border formation. All cell lines were tested for mycoplasma contamination prior to use in this work.

## Immunofluorescence, mCLING treatment, and antibody staining

Cells were fixed for 10 min with prewarmed (37°C) 4% paraformaldehyde + 0.1% glutaraldehyde in PBS. For visualization of lipid membranes, cells were washed twice in PBS after fixation and incubated in 5 µM either mCLING-Atto647N (Synaptic Systems, 710 006AT1) or mCLING-Atto488 (Synaptic Systems, 710 006AT3) in PBS overnight at RT. The following day, cells were fixed a second time with prewarmed (37°C) 4% paraformaldehyde + 0.1% glutaraldehyde in PBS. For clathrin heavy chain labeling in *Figure 3E* and tubulin labeling of COS7 cells in *Figure 3G*, cells were pre-extracted for 1 min with prewarmed (37°C) extraction buffer (80 mM K-PIPES pH 6.8, 4 mM MgCl$_2$, 1 mM EGTA, 0.35% Trition X-100, 0.2% glutaraldehyde). After extraction, cells were fixed for 10 min with prewarmed 4% paraformaldehyde in PBS. Next, cells were washed with PBS and permeabilized using PBS + 0.2% Triton X-100. Cells should be permeabilized even if no antibody staining will be done to ensure uniform gelation. Epitope blocking and antibody labeling steps were performed in PBS + 3% BSA. For immunofluorescence staining, we used a rabbit monoclonal antibody against α-tubulin (clone EP1332Y, Abcam, ab52866), a mouse monoclonal antibody against clathrin heavy chain (Thermo Fisher Scientific, MA1-065), and a chicken polyclonal antibody against GFP (Aves Labs, GFP-1010) in combination with goat anti-rabbit IgG (H+L) Alexa Fluor 594 (Molecular Probes, a11037), goat anti-mouse IgG (H+L) Alexa Fluor 488 (Molecular Probes, a11029), and goat anti-chicken IgY (H+L) Alexa Fluor 488 (Molecular Probes, a11039), respectively.

## TREx

For TREx experiments shown in *Figures 3E–G*, *4 and 5*, samples were treated with 100 µg/mL acryloyl-X SE (AcX) (Thermo Fisher, A20770) in PBS overnight at RT. In these experiments, TEMED and bis were used at a concentration of 15 ppt and 90 ppm, respectively. 170 µL of gelation solution was transferred to a silicone gasket with inner diameter of 13 mm (Sigma-Aldrich, GBL664107) attached to a parafilm-covered glass slide, with the sample put cell-down on top to close off the gelation chamber. The sample was directly transferred to a 37°C incubator for 1 hr to fully polymerize the gel. All gels excluding samples that were processed for subsequent NHS ester staining were transferred to a 12-well plate and digested with 7.5 U/mL Proteinase-K (Thermo Fisher, EO0491) in TAE buffer (containing 40 mM Tris, 20 mM acetic acid, and 1 mM EDTA) supplemented with 0.5% Triton X-100, 0.8 M guanidine-HCl, and DAPI for 4 hr at 37°C. The gel was transferred to a Petri dish, water was exchanged 2 × 30 min, and the sample was left in MilliQ water to expand overnight.

For NHS staining, gels were first treated in disruption buffer containing 200 mM SDS, 200 mM NaCl, and 50 mM Tris pH 6.8 for 1.5 hr at 78°C. Gels were washed twice for 15 min in PBS and incubated with 20 µg/mL Atto 594 NHS ester (Sigma-Aldrich, 08471) in PBS prepared from a 20 mg/mL stock solution in DMSO for 1 hr at RT with shaking. After staining, gels were washed with excess of PBS, transferred to a Petri dish, and expanded overnight. Prior to imaging, the cells were trimmed using a scalpel blade to fit in an Attofluor Cell Chamber (Molecular Probes A-7816).

## Image acquisition and analysis

ExM and pre-expansion images were acquired using a Leica TCS SP8 STED 3X microscope equipped with an HC PL APO ×86/1.20W motCORR STED (Leica 15506333) water objective. A pulsed white laser (80 MHz) was used for excitation; when using STED, a 775 nm pulsed depletion laser was used. The internal Leica GaAsP HyD hybrid detectors were used with a time gate of $1 \leq tg \leq 6$ ns. The set-up was controlled using LAS X.

All data processing and analysis were done using MATLAB, ImageJ, and Arivis.

*Figure 3E* panels are sum projections of three planes (z-spacing 0.35 µm). For the CCP diameter analysis in *Figure 3F*, line scans over individual CCPs that had clearly distinguishable central nulls were drawn. From these line scans, the peak-to-peak distance was determined, which corresponds to the diameter of each CCP. All line scans were generated using ImageJ and processed using GraphPad Prism 8. *Figure 3H* panels are maximum intensity projections of the bottom ~1 µm of cells. For *Figure 3I and J*, BigWarp (*Bogovic et al., 2016*) was used to manually pick control points for nonrigid registration. The analysis scripts 'bigwarpSimilarityPart.groovy' and 'Apply_Bigwarp_Xfm_csvPts.groovy' were used to calculate deformation fields that register expanded images to pre-expansion images and decompose each deformation field into a similarity part (corresponding to theoretical ideal expansion) and a residual elastic part (thin-plate spline, corresponding to nonideal deformations

introduced by expansion), adapted from *Jurriens et al., 2021*. The similarity part was used to find the macroscopic expansion factor, while the residual elastic part was used to calculate the measurement error as follows. A MATLAB script was used to calculate the measurement error for all pairs of points in the image as described in *Chen et al., 2015* by finding the magnitude of the difference between the displacement vectors for each pair of points in the residual elastic deformation field. These differences were binned according to the distance between points in the pre-expansion image. For each measurement length bin, the mean and standard deviation of measurement errors were calculated and plotted.

*Figure 4A* raw data was imported in Arivis, a Discrete Gaussian Filter with smoothing radius of 2 was applied, and this dataset was used for volumetric renders and clipping. Gamma was adjusted manually to increase visibility of plasma membrane ruffles and intracellular organelles in the same view. For *Figure 4B*, the same raw dataset was imported in ImageJ and a sum projection of three planes (z-spacing: 0.35 µm) around the plane of the immunological synapse was segmented for mitochondria using the trainable Weka segmentation plugin in ImageJ. *Figure 4C* is a sum projection of 3 planes (z-spacing 0.35 µm). The line scan in *Figure 4C* was generated using ImageJ and processed using GraphPad Prism 8. For *Figure 4D*, raw data was imported in Arivis, a Discrete Gaussian Filter with smoothing radius of 2 was applied, and this dataset was volumetrically rendered with the opacity mapped to the z-axis. *Figure 4E* is a sum projection of five slices (z-spacing 0.35 µm). *Figure 4F–H* are sum projections of three planes (z-spacing: 0.35 µm) and respective zooms. For the MV diameter analysis in *Figure 4H*, sum projections of three planes were thresholded (ImageJ, set to auto), watershed to split joining particles, and the area determined using the analyze particles function in ImageJ, which was converted to diameter as in *Julio et al., 2008*.

*Figure 5A* panels are sum projections of three planes (z-spacing before expansion and after expansion 0.07 and 0.15 µm, respectively), reslices are sum projections (three planes) of resliced data. *Figure 5B* panels are sum projections of three planes (z-spacing: 0.35 µm). For ER tubule diameter analysis in *Figure 5C*, line scans were drawn over tubules that were positive for both Sec61-GFP and mCLING. The full width at half maximum was determined for each line scan. For *Figure 5D*, raw data was imported into Arivis, a Discrete Gaussian Filter with smoothing radius of 2 was applied, and this dataset was used for volumetric renders and clipping. Shown single planes are sum projections of three slices (z-spacing 0.35 µm) of the same raw data and were processed using ImageJ. *Figure 5E* is a maximum projection of three planes (z-spacing: 0.35 µm).

## Acknowledgements

We are grateful to Sven van IJzendoorn (UMCG) and Wilco Nijenhuis (UU) for providing the Caco2 monolayer samples. We thank the Janelia light microscopy core facility for the use of confocal and lightsheet microscopes. We thank the Janelia cell culture core facility for maintaining and providing cultured cells. We thank the Janelia histology core facility for providing tissue slices. AA was supported by the Netherlands Organization for Scientific Research Spinoza Prize. LCK was supported by the European Research Council (ERC Consolidator Grant 819219). BM, ME, and PWT were supported by the Howard Hughes Medical Institute (HHMI).

## Additional information

### Funding

| Funder | Grant reference number | Author |
| --- | --- | --- |
| Netherlands Organization for Scientific Research | Spinoza Prize | Anna Akhmanova |
| European Research Council | ERC Consolidator Grant 819219 | Lukas C Kapitein |
| Howard Hughes Medical Institute | | Boaz Mohar<br>Mark Eddison<br>Paul W Tillberg |

| Funder | Grant reference number | Author |
|---|---|---|

The funders had no role in study design, data collection and interpretation, or the decision to submit the work for publication.

## Author contributions

Hugo GJ Damstra, Formal analysis, Investigation, Methodology, Validation, Visualization, Writing – original draft, Writing – review and editing; Boaz Mohar, Formal analysis, Investigation, Methodology, Visualization, Writing – original draft, Writing – review and editing; Mark Eddison, Formal analysis, Investigation, Methodology, Writing – review and editing; Anna Akhmanova, Funding acquisition, Supervision, Writing – review and editing; Lukas C Kapitein, Conceptualization, Formal analysis, Funding acquisition, Investigation, Project administration, Supervision, Visualization, Writing – original draft, Writing – review and editing; Paul W Tillberg, Conceptualization, Formal analysis, Investigation, Methodology, Resources, Software, Supervision, Validation, Writing – original draft, Writing – review and editing

## Author ORCIDs

Hugo GJ Damstra (iD) http://orcid.org/0000-0003-0847-609X
Boaz Mohar (iD) http://orcid.org/0000-0002-8613-2869
Anna Akhmanova (iD) http://orcid.org/0000-0002-9048-8614
Lukas C Kapitein (iD) http://orcid.org/0000-0001-9418-6739
Paul W Tillberg (iD) http://orcid.org/0000-0002-2568-2365

## Decision letter and Author response

Decision letter https://doi.org/10.7554/eLife.73775.sa1
Author response https://doi.org/10.7554/eLife.73775.sa2

# Additional files

## Supplementary files

• Transparent reporting form

## Data availability

All data generated or analyzed during this study are included in the manuscript and supporting file; Source data files are available at https://doi.org/10.6084/m9.figshare.19142264.

The following dataset was generated:

| Author(s) | Year | Dataset title | Dataset URL | Database and Identifier |
|---|---|---|---|---|
| Damstra HGJ, Mohar B, Eddison M, Akhmanova A, Kaptein LC, Tillberg P | 2022 | Source data for paper: Visualizing cellular and tissue ultrastructure using Ten-fold Robust Expansion Microscopy (TREx) | https://doi.org/10.6084/m9.figshare.19142264 | figshare, 10.6084/m9.figshare.19142264 |

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
