## [Editor Report]

The new robust Ten-fold Robust Expansion Microscopy method developed by the authors should be of wide interest to the cell biology community.

---

## [Decision Letter]

**Decision letter after peer review:**

Thank you for submitting your article "Visualizing cellular and tissue ultrastructure using Ten-fold Robust Expansion Microscopy (TREx)" for consideration by *eLife*. Your article has been reviewed by 3 peer reviewers, and the evaluation has been overseen by a Reviewing Editor and Suzanne Pfeffer as the Senior Editor. The following individual involved in review of your submission has agreed to reveal their identity: Xiaoyu Shi (Reviewer #2).

Essential revisions:

All reviewers agreed that this is an interesting and substantial advance in the field of expansion microscopy to merit publication in *eLife*. The reviewers made constructive suggestions about how the manuscript can be improved with further discussion/clarification, some additional analysis. In particular:

1) Multiple reviewers felt that the paper would benefit from showing more zoomed-in views of many of the images to better demonstrate the enhanced resolution and quality of the images.

2) There were multiple questions about the use of NHS-dyes, justification for the specific choice of the NHS-dye or demonstration of whether other NHS-dyes are appropriate. In addition, questions arose about overlap of NHS-dyes with mCLING.

3) A better quantification of the local expansion factors for various organelles will improve the paper's impact.

4) Please note that during the consultation session, Reviewer 1 noted the following in response to the first comment of Reviewer 3 regarding distortion:

Reviewer #3 pointed out that the microtubule distortions seem larger than earlier published values from Chen 2015 or Tillberg 2016. "I hadn't inspected that closely in my initial review, so I took a moment to do so now.

In Chen 2015 and Tillberg 2016 the distortion plots rise quickly and sort of plateau, whereas in Damstra the distortion plot is sort of linear. So comparing the older and newer results will depend on where you look. At longer distances like 20um, Damstra has ~750nm RMS error, Chen 2015 has ~250nm RMS error, and Tillberg 2015 has ~220nm RMS error. At shorter distances like 2.5um, Damstra has around 100nm RMS error, Chen 2015 has ~120nm RMS error, and Tillberg 2016 has ~90nm RMS error.

My take on this is the following. Damstra shows comparable RMS error at shorter distances (arguably the more important regime for a super-resolution technique) but more distortion at longer distances. It is not particularly concerning to me, but Damstra could consider commenting on this as suggested by reviewer #3 including mentioning whether the 42 fields of view used to create the distortion plot were from a single gel/preparation or from multiple gels/preparations. "

*Reviewer #2 (Recommendations for the authors):*

Line 179-181 and Methods – Synaptic distance: Please clarify if the distance between the pre- and post-synapses are measured in 3D images or 2D projection. The 3D images should provide more accurate measurement. The authors should measure the 3D distance.

Line 378: Why had the authors repeatedly failed to cleanly recover epithelial cultures using standard ExM?

*Reviewer #3 (Recommendations for the authors):*

1) The measurement error due to distortions induced by the gel expansion process was quantified using tubulin imaging, which is an ideal way of measuring this issue. However, while the authors indicate that the error "is in line with previous expansion methods (F. Chen et al., 2015; Tillberg et al., 2016)", this does not seem to be precisely the case. The error seems almost 10-fold greater than the Chen et al., 2015 paper, and 4-5 fold greater than in the Tillberg et al., 2016 paper. Also, it is unclear how representative the error shown in Figure 3G is. How many images, from different fields and gels, were analyzed here?

2) It seems like many of the images show low-zoom views of the expanded samples, where little can be seen in terms of details. How continuous are the microtubules shown in Figure 3? High-zoom views would be useful.

3) Similar to the previous comment, there is surprisingly little detail to be seen in the active zones and postsynaptic densities from Figure 2B-C. In principle, with 10-fold expansion one should be able to observe nanomodules or domains within both the presynaptic active zone and postsynaptic density (e.g. Hruska et al., Nat Neurosci, 2018). Could the image acquisition be improved, to enable the description of such synaptic elements?

4) The issue of optimized imaging applies also to the nuclear pore images in Figure 3. The type of image provided by the authors is superior to the 4x-expansion images provided in previous publications (e.g. Thevathasan et al., Nat Methods, 2019), but does not reach the quality of previous STED or STORM images of these structures (see the same publication as mentioned above), albeit the resolution enhancement should be higher in the current manuscript. Could the authors use a higher magnification objective, thereby showcasing the entire potential of their technology?

---

## [Author Response]

Essential revisions:All reviewers agreed that this is an interesting and substantial advance in the field of expansion microscopy to merit publication in eLife. The reviewers made constructive suggestions about how the manuscript can be improved with further discussion/clarification, some additional analysis. In particular:1) Multiple reviewers felt that the paper would benefit from showing more zoomed-in views of many of the images to better demonstrate the enhanced resolution and quality of the images.

We have included more zoomed-in high resolution views of microtubules, microtubules interacting with mitochondria and ER, close ER associations with mitochondria and other membranous organelles, close association of clathrin coated pits and microtubules, zooms of nuclear pore complexes, and zooms of synapses in mouse brains showing sub-synaptic domains of Bassoon and Homer (additions can be found in Figure 3, 5 and Figure 2 – Figure Supp. 1).

2) There were multiple questions about the use of NHS-dyes, justification for the specific choice of the NHS-dye or demonstration of whether other NHS-dyes are appropriate. In addition, questions arose about overlap of NHS-dyes with mCLING.

For cultured cells, we used AlexaFluor594-NHS (which is spectrally similar to Atto594-NHS, demonstrated extensively in pan-ExM^1^) and for tissue we used the inexpensive, green emitting BODIPY-FL-NHS. The reaction mechanism of the NHS-ester moiety with primary amines should not be affected by the identity of the dye attached to the NHS, but according to Sim et al.,^2^ the reaction efficiency is apparently modulated by non-covalent interactions between the dye and its protein targets in a way that is mainly influenced by the hydrophobicity of the dye. We have updated the Discussion section to reflect this information from the literature, and how it relates to our dyes, Alexa594 and BODIPY-FL:

“Sim et al. (Sim et al., 2021) have shown that highly hydrophobic NHS ester dyes exhibit strong contrast for cytosolic organelles while highly hydrophilic NHS ester dyes strongly stain the nucleus. The moderate hydrophobicity dyes that we used (BODIPY-FL (Zanetti-Domingues, Tynan, Rolfe, Clarke, and Martin-Fernandez, 2013) and AlexaFluor594 (Hughes, Rawle, and Boxer, 2014)) exhibit both nuclear staining and contrast for cytosolic organelles.”

3) A better quantification of the local expansion factors for various organelles will improve the paper's impact.

We have further validated our method by quantifying the width of ER tubules for both antibody labeling and direct labeling by mCLING intercalation. The ER tubule diameter obtained falls within the previously published range, and we show that we have sufficient resolution to visualize the effect of linkage error on ER tubule diameter (i.e. difference between mCLING and indirect immuno-labeled Sec61-GFP). Moreover, we have included a quantification of clathrin-coated pit (CCP) diameter and microtubule M-profiles, two other benchmarkes for super resolution microscopy. These additions can be found in Figure 3 and Figure 5.

4) Please note that during the consultation session, Reviewer 1 noted the following in response to the first comment of Reviewer 3 regarding distortion:Reviewer #3 pointed out that the microtubule distortions seem larger than earlier published values from Chen 2015 or Tillberg 2016. "I hadn't inspected that closely in my initial review, so I took a moment to do so now.In Chen 2015 and Tillberg 2016 the distortion plots rise quickly and sort of plateau, whereas in Damstra the distortion plot is sort of linear. So comparing the older and newer results will depend on where you look. At longer distances like 20um, Damstra has ~750nm RMS error, Chen 2015 has ~250nm RMS error, and Tillberg 2015 has ~220nm RMS error. At shorter distances like 2.5um, Damstra has around 100nm RMS error, Chen 2015 has ~120nm RMS error, and Tillberg 2016 has ~90nm RMS error.My take on this is the following. Damstra shows comparable RMS error at shorter distances (arguably the more important regime for a super-resolution technique) but more distortion at longer distances. It is not particularly concerning to me, but Damstra could consider commenting on this as suggested by reviewer #3 including mentioning whether the 42 fields of view used to create the distortion plot were from a single gel/preparation or from multiple gels/preparations. "

We have clarified the text to reflect that the fractional error over short distances is comparable to previous findings with expansion methods, while the absolute error does not saturate until greater distances in the case of TREx, and that the 42 fields of view are tiles of one image from one specimen. We agree with reviewer #1, that for the vast majority of applications, the fidelity over sub-cellular distances is more important than over large distances.

Concerning the long-range distortions, these could come from two sources—gel handling (extrinsic) and gel heterogeneity (intrinsic). Distortions of the first type could be due to stresses applied while mounting the gel on poly-L-lysine coated glass surfaces. We have attempted to disentangle these contributions by studying a case combining both worse-case and best-case extrinsic distortion. We expanded a large (2.5x5mm) portion of cultured U2OS cells (Author response image 1, just gelled, before disruption/expansion) and instead of immobilizing the gel by sticking it down with poly-lysine, we simply allowed it to rest with its short side in contact with the side of a petri dish (Author response image 1, after expansion). In this case, the side against the wall experiences maximum extrinsic distortion while the other side, far away from this stress, should experience minimum extrinsic distortion. We carried out nonrigid registration from the post- to the pre-expansion image and found that indeed the distortion was greatest at the bottom (Petri dish-contacting) edge (Author response image 1 , deformation field). The measurement error calculated over the entire gel was a non-saturating proportional error on the order of 5% (Author response image 1, measurement error). The deformation at the edge appears to propagate up to several mm along the length of the gel.

**Author response image 1. sa2fig1:** Quantification of gel deformation over long distances. (A) Cultured U2OS cells just after gelation and (B) after full expansion, with an FFT-based short pass filter applied to reduce variation in illumination across imaging fields. (C) Map of distortions across the entire gel. (D) Average measurement error across the entire gel (mean ± SD in blue) and (E) measurement error (mean) of 1,000-pixel square areas indicated by colored boxes in (C) at increasing distances away from the free gel edge. Scale bars: (A) 1 mm, (B) 10 mm (post-expansion), (C) 1 mm (pre-expansion).

We calculated the measurement error for restricted fields of view (1,000 pix = 650 µm square) at varying positions relative to the free edge of the gel (indicated in boxes on Author response image 1). At the free edge, errors saturate at ~1 µm (Author response image 1, blue), while closer to the distorted edge errors more closely resemble the non-saturating result found for the whole gel (Author response image 1 , red and yellow). Thus, the measurement error profile varies from point to point in the gel, and extreme extrinsic distortions can propagate far away from the source of the distortion. For the rare applications where nanoscale precision over macroscopic distances is required, more development will be required to minimize these extrinsic errors by optimizing gel handling and mounting, but this is beyond the scope of the present work. These considerations were added to the discussion in the paper.

Finally, in the course of doing these follow-up experiments, we discovered that omitting permeabilization prior to gel embedding (i.e. when not doing such a step during immunostaining) leads to sporadic failures. We now include in the text that permeabilization must be included even if there is no antibody staining.

Reviewer #2 (Recommendations for the authors):Line 179-181 and Methods – Synaptic distance: Please clarify if the distance between the pre- and post-synapses are measured in 3D images or 2D projection. The 3D images should provide more accurate measurement. The authors should measure the 3D distance.

We have measured these in 3D, the text has been edited to clarify.

Line 378: Why had the authors repeatedly failed to cleanly recover epithelial cultures using standard ExM?

We now address this point briefly in the discussion. In short, we speculate the increased robustness of TREx is less affected by interference from the silicone filter on the gelation process, likely due to the increased monomer content.

Reviewer #3 (Recommendations for the authors):1) The measurement error due to distortions induced by the gel expansion process was quantified using tubulin imaging, which is an ideal way of measuring this issue. However, while the authors indicate that the error "is in line with previous expansion methods (F. Chen et al., 2015; Tillberg et al., 2016)", this does not seem to be precisely the case. The error seems almost 10-fold greater than the Chen et al., 2015 paper, and 4-5 fold greater than in the Tillberg et al., 2016 paper. Also, it is unclear how representative the error shown in Figure 3G is. How many images, from different fields and gels, were analyzed here?

We address this in modifications to the text and new data described in detail in response to essential point 4, above.

2) It seems like many of the images show low-zoom views of the expanded samples, where little can be seen in terms of details. How continuous are the microtubules shown in Figure 3? High-zoom views would be useful.

We address this in our response to essential point 1, above.

3) Similar to the previous comment, there is surprisingly little detail to be seen in the active zones and postsynaptic densities from Figure 2B-C. In principle, with 10-fold expansion one should be able to observe nanomodules or domains within both the presynaptic active zone and postsynaptic density (e.g. Hruska et al., Nat Neurosci, 2018). Could the image acquisition be improved, to enable the description of such synaptic elements?

Additional insets with line profiles across synapses were added to Figure 2 – Supp. Figure 1b to illustrate example synapses with non-uniform distribution of the PSD and its pre-synaptic active zone.

4) The issue of optimized imaging applies also to the nuclear pore images in Figure 3. The type of image provided by the authors is superior to the 4x-expansion images provided in previous publications (e.g. Thevathasan et al., Nat Methods, 2019), but does not reach the quality of previous STED or STORM images of these structures (see the same publication as mentioned above), albeit the resolution enhancement should be higher in the current manuscript. Could the authors use a higher magnification objective, thereby showcasing the entire potential of their technology?

The limiting factor on NPC image quality appears to be the density of signal retention rather than the imaging resolution. We agree that this is an important issue for the field to address, but it would not be improved by increasing the resolution of imaging and is outside the scope of the present work.

References

1. M’Saad, O. and Bewersdorf, J. Light microscopy of proteins in their ultrastructural context. *Nat. Commun.* (2020). doi:10.1038/s41467-020-17523-8

2. Sim, J. *et al.* Whole-ExM: Expansion microscopy imaging of all anatomical structures of whole larval zebrafish. *bioRxiv* 2021.05.18.443629 (2021).

3. Zanetti-Domingues, L. C., Tynan, C. J., Rolfe, D. J., Clarke, D. T. and Martin-Fernandez, M. Hydrophobic Fluorescent Probes Introduce Artifacts into Single Molecule Tracking Experiments Due to Non-Specific Binding. *PLoS One*
**8**, 74200 (2013).

4. Hughes, L. D., Rawle, R. J. and Boxer, S. G. Choose your label wisely: Water-soluble fluorophores often interact with lipid bilayers. *PLoS One*
**9**, (2014).